# RELIABLE AND DIVERSE EVALUATION OF LLM MEDICAL KNOWLEDGE MASTERY

**Yuxuan Zhou**[1]**, Xien Liu**[*,1]**, Chen Ning**[1]**, Xiao Zhang**[1]**, Ji Wu**[1,2,3]
[1]Department of Electronic Engineering, Tsinghua University, Beijing, China
[2]College of AI, Tsinghua University, Beijing, China    [3]BNRist, Beijing, China

## ABSTRACT

Mastering medical knowledge is crucial for medical-specific LLMs. However, despite the existence of medical benchmarks like MedQA, a unified framework that fully leverages existing knowledge bases to evaluate LLMs' mastery of medical knowledge is still lacking. We propose PretexEval, a novel framework that dynamically generates reliable and diverse test samples to evaluate LLMs for any given medical knowledge base. We notice that test samples produced directly from knowledge bases by templates or LLMs may introduce factual errors and also lack diversity. To address these issues, our framework employs predicate equivalence transformations to produce a series of variants for any given medical knowledge point. Finally, these produced predicate variants are converted into textual language, resulting in a series of reliable and diverse test samples. Here, we use our proposed framework to systematically investigate the mastery of medical factual knowledge of 12 well-known LLMs, based on two knowledge bases that are crucial for clinical diagnosis and treatment. The evaluation results illustrate that current LLMs still exhibit significant deficiencies in fully mastering medical knowledge, despite achieving considerable success on some famous public benchmarks. These new findings provide valuable insights for developing medical-specific LLMs, highlighting that current LLMs urgently need to strengthen their comprehensive and in-depth mastery of medical knowledge before being applied to real-world medical scenarios.

## 1 INTRODUCTION

Recent years have witnessed the rapid advancement of large language models (LLMs), which have exhibited potential across various domains (Brown et al., 2020; Ouyang et al., 2022; Touvron et al., 2023; OpenAI, 2023; Madani et al., 2023; Boiko et al., 2023), including medicine. Solving medical problems requires LLMs to master medical factual knowledge comprehensively and in-depth. Recent studies (Singhal et al., 2023; Nori et al., 2023; Pal & Sankarasubbu, 2024) showed that some LLMs (e.g. GPT-4) encode medical factual knowledge, achieving outstanding performance across multiple medical benchmarks (Jin et al., 2019; 2021; Pal et al., 2022; Singhal et al., 2023; Sung et al., 2021; Meng et al., 2022), such as MedQA. Constructed through expert annotation, these benchmarks have long been effective tools for evaluating LLMs' medical capabilities. However, they may face challenges such as becoming outdated or being possibly leaked to LLMs, which could lead to evaluations that lack reliability. Meanwhile, medical databases such as UMLS (Bodenreider, 2004) are regularly updated and contain extensive medical knowledge, but there is currently no unified framework that fully leverages these knowledge bases to evaluate LLMs' mastery of medical knowledge. Therefore, we aim to bridge this gap in this study by proposing an evaluation framework that investigates LLMs' medical knowledge mastery based on any given medical knowledge base.

Evaluating LLMs using medical knowledge bases requires generating textual test samples from structured knowledge. A straightforward method is to prompt LLMs to directly generate test samples based on specific knowledge points. However, this method has two drawbacks as illustrated in Figure 1: (1) **insufficient factuality**: factual errors (e.g. incorrect relations) may be introduced during LLM generation process, affecting the reliability of evaluation; and (2) **low structure diversity**: samples

---

[*]Corresponding author.

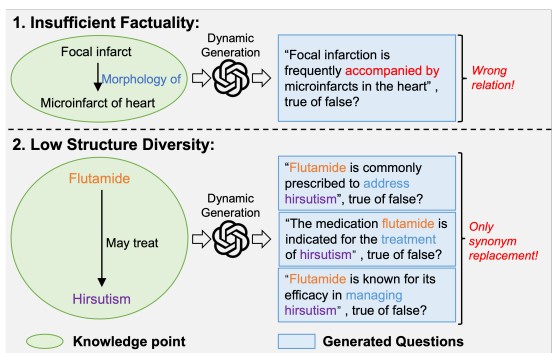

Figure 1: Drawbacks of test samples produced directly by LLMs: (1) LLMs may introduce factual errors into generated samples; (2) Samples directly generated by LLMs exhibit low diversity.

Figure 2: Schema of the proposed Predicate-to-text evaluation method (Top) compared with directly generating test variants by LLMs (Bottom).

generated from the same knowledge point differ primarily in wording (e.g. synonym replacement) rather than in expression structure, compromising the diversity of evaluation.

The purpose of this study is to develop a unified evaluation framework that dynamically generates reliable and diverse test samples from medical knowledge bases to probe LLMs' mastery of medical factual knowledge. Given that medical factual knowledge primarily involves relationships between medical entities, it can be effectively expressed through predicates. Inspired by this, we propose a **Pre**dicate-to-**tex**t **Eval**uation method (**PretexEval**) that dynamically generates reliable and structurally diverse test samples based on the medical knowledge points from knowledge bases. Figure 2 presents the schema of our method. Specifically, we first express each knowledge point using a predicate expression. Then, we derive a series of predicate variants from this predicate expression through predicate equivalent/implication transformation[1]. Such predicate transformation process enhances the structural diversity of generated test samples and also effectively prevents the introduction of factual errors. Finally, we use a prototype-based method to convert these variants back into the textual space to create the test samples. This method first transforms the predicate variants into prototype samples using templates to ensure reliability, and then rephrases these prototypes with LLMs to enhance the readability and lexical diversity of the generated samples. It is worth noting that the proposed evaluation method is highly **versatile** and can be applied to any medical knowledge base with minimal adjustments to evaluate LLMs' mastery of the knowledge it contains.

In our study, we employ the proposed evaluation framework to systematically investigate the mastery of medical knowledge among 12 well-known LLMs, using two medical knowledge bases that contain essential information for clinical diagnosis and treatment. Experimental results indicate that the performance of current LLMs on test samples generated by our method is significantly lower than on samples directly generated by handcrafted templates or by prompting an LLM. Furthermore, these evaluated LLMs exhibit notable inconsistency in handling test samples derived from the same knowledge point. These findings indicate that, despite their impressive performance on several medical benchmarks, current LLMs have not fully mastered the medical knowledge essential for real-world clinical tasks. Therefore, they may require additional training before being applied in real-world medical scenarios to further enhance their mastery of medical knowledge. We release the codes and datasets to facilitate future study[2]. Our contributions are summarized as follows:

- We introduce PretexEval, a predicate-to-text method that dynamically generates reliable and structurally diverse test samples based on any given medical knowledge base.

- Using the proposed method, we systematically investigate the medical factual knowledge mastery of 12 well-known LLMs based on two knowledge bases closely related to clinical diagnosis and treatment.

---

[1]For the sake of convenience, we refer to both equivalent and implication transformations collectively as predicate equivalent transformations in the following sections, without making a distinction between them.

[2]https://github.com/THUMLP/PretexEval

- Our findings reveal that current LLMs have not yet comprehensively and deeply mastered medical knowledge, underscoring the urgent need to improve their medical knowledge mastery before applying them to real-world medical tasks.

## 2 RELATED WORK

**LLM Medical Evaluation**  Current medical evaluation benchmarks for LLMs can be divided into two categories: (1) QA datasets that evaluate LLMs' comprehensive medical capabilities with questions collected from medical literature (Jin et al., 2019), exams (Jin et al., 2021; Pal et al., 2022), or online websites (Singhal et al., 2023); (2) datasets for testing the mastery of LLM medical knowledge (Sung et al., 2021; Meng et al., 2022). These static benchmarks are meticulously created by medical experts and possess high reliability. However, they may face problems such as becoming outdated or leaked to LLMs, affecting the comprehensiveness of evaluation. While constructing new benchmarks can alleviate these problems, they will also become obsolete over time.

**Dynamic Evaluation Schema**  Several studies have proposed dynamic evaluation methods that automatically generate new test samples, effectively addressing the issue of unreliable evaluation caused by LLMs memorizing test samples (benchmark leakage). Some works leverage algorithms to dynamically generate test samples for specific tasks, such as mathematics (Zhu et al., 2024a) and SQL execution (Lei et al., 2023). Others (Zhu et al., 2024c;b) generate test samples by paraphrasing existing benchmarks. However, currently there is no related work that uses dynamic evaluation methods to evaluate LLMs based on knowledge bases. By dynamically generating test samples from regularly updated medical knowledge bases, our framework can effectively address issues associated with static evaluation benchmarks (becoming outdated, being leaked to LLMs).

## 3 METHOD

### 3.1 EVALUATION SCHEMA

In this section, we introduce the schema of our PretexEval method, which generates structural diverse and reliable test samples for LLM factual knowledge evaluation. Given a knowledge point P, a straightforward idea is to directly generate a test sample using an LLM:

$$S = G_{LLM}(P) \tag{1}$$

where $G_{LLM}$ denotes the LLM generation process, and S refers to the generated test sample. As introduced above, $G_{LLM}$ may create samples that lack diversity and reliability. In contrast, our method first expresses the knowledge point using a predicate expression and then derives a series of variants via predicate equivalent transformation:

$$q = T_{text2pre}(P) \tag{2}$$

$$[v_1, v_2, \cdots, v_m] = T_{Eq}(q) \tag{3}$$

where $T_{text2pre}$ denotes a mapping that projects the original knowledge point $P$ into the predicate expression q. $T_{Eq}$ refers to the predicate equivalent transformation, and $\{v_i\}_{i=1}^m$ are the variants derived from the original expression q. The property of predicate equivalent transformation ensures the reliability of these variants, provided that the original expression q is true:

$$(q = \text{True}) \Rightarrow (v_i = \text{True}), \quad 1 \le i \le m \tag{4}$$

Finally, we convert each predicate variant back to a textual test sample for evaluation:

$$S_i = T_{pre2text}(v_i), \quad 1 \le i \le m \tag{5}$$

where $T_{pre2text}$ maps each predicate variant $v_i$ into a corresponding test sample (textual variant). Since these samples are derived from predicate variants with diverse structures, the predicate-text duality ensures they exhibit substantial diversity while maintaining reliability.

### 3.2 EVALUATION FRAMEWORK

Building on the proposed evaluation schema, we develop a novel evaluation framework to comprehensively evaluate LLMs' mastery of medical factual knowledge. Figure 3 presents an overview of this framework.

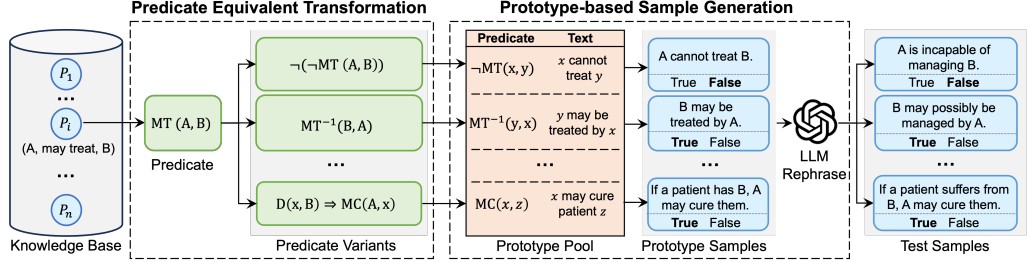

Figure 3: An overview of the proposed PretexEval framework, which dynamically generates test samples from any medical knowledge base for evaluating LLMs' medical knowledge mastery.

### 3.2.1 PREDICATE VARIANT GENERATION

A single knowledge point (i.e. knowledge triplet in knowledge bases) can be denoted as $P = (h, r, t)$, where $h$, $r$, and $t$ refer to the head entity, the relation, and the tail entity, respectively. In predicate logic, such a relation can be effectively presented by:

$$q = \mathcal{R}(h, t) \tag{6}$$

where $\mathcal{R}(x, y)$ is a predicate derived from the relation $r$, representing the statement "x has the relation r with y", where $q$ represents its value at the point $(h, t)$. Next, the framework employs three types of equivalent transformations that are widely employed in practical medical applications, including:

- **Inversion** ($\mathcal{R}^{-1}(t, h)$): The inverse expression presents the original expression from another direction. For example, if the statement "Drug A may treat disease B" holds, then "Disease B may be treated by drug A" also holds.

- **Instantiation** ($\mathcal{P}(h, x) \Rightarrow \mathcal{Q}(x, t)$): This type of transformation applies a general knowledge point to a specific case. For example, the statement "Drug A may treat disease B" can be instantiated as "If a patient has disease B, drug A may cure them." Such transformation is commonly used in disease diagnosis and treatment.

- **Double Negation** ($\neg(\neg \mathcal{R}(h, t))$): The double negation rule is widely utilized to obtain logically equivalent expressions. In our framework, this rule is applied to construct **negative** expressions. For example, if "Drug A may treat disease B" holds, then "Drug A cannot treat disease B" must be false.

It is noteworthy that these three types of transformation can be further combined to produce additional expressions based on the transitive property of predicate equivalent transformation. As a result, a total of $m$ variants are generated in this process:

$$v_i = T_{Eq}^i(\mathcal{R}(h, t)), \quad 1 \leq i \leq m \tag{7}$$

where $T_{Eq}^i$ denotes the $i^{th}$ predicate equivalent transformation.

### 3.2.2 TEXTUAL SAMPLE GENERATION

Finally, the predicate variants are converted back into textual samples for LLM evaluation. A straightforward method is template-based rephrasing, but the resulting samples may lack fluency, potentially impacting the model's performance. Another approach is to prompt LLMs to generate test samples directly from predicates. However, considering that LLMs may not fully understand predicate forms, this method can also introduce factual errors. To address this issue, we designed a prototype-based sample generation method. Specifically, for each predicate variant $T_{Eq}^i(\mathcal{R}(h, t))$, we initially retrieve the corresponding prototype from a pre-constructed prototype pool based on the predicate $T_{Eq}^i \cdot \mathcal{R}$. For example, considering the predicate variant $MT^{-1}(y, x)$ in Figure 3, we retrieve the corresponding prototype "$y$ may be treated by $x$". For predicate variants obtained through double negation, we retrieve prototypes based on their negated form (i.e., single negation form) to

generate **negated samples** for LLM evaluation. Subsequently, the prototype is instantiated by the arguments $(h, t)$. The instantiated prototype precisely conveys the predicate variant in the textual space. Finally, the prototype is further rephrased by an LLM to obtain the final test sample $S_i$. Since current LLMs possess strong language capabilities and seldom make mistakes in sentence rephrasing, the proposed sample generation method can ensure the reliability and diversity of the generated samples.

### 3.2.3 EVALUATION METRICS

In our framework, we mainly evaluate LLMs using statement verification tasks, asking them to determine whether a given statement is true or false:

$$\text{Score}(M, S_i) = \mathbb{1}(M(S_i) = l_i), 1 \leq i \leq m \tag{8}$$

where $M$ is the evaluated LLM, $S_i$ is the textual variant (statement) generated by our framework, and $M(S_i) \in \{\mathbf{T}, \mathbf{F}\}$ denotes LLM's prediction for $S_i$. $l_i \in \{\mathbf{T}, \mathbf{F}\}$ is the label of $S_i$, and the function $\mathbb{1}(\cdot)$ is a characteristic function that equals 1 when the enclosed expression is true, and 0 otherwise. Additionally, we also adopt 4-option multiple-choice questions to further validate the scalability of PretexEval on various evaluation tasks (see Section 4.2.4).

For a dataset with $n$ knowledge points $\{P_j\}_{j=1}^n$, we initially use the metric *average accuracy* to compute the accuracy across all test samples:

$$a_{\text{avg}} = \frac{1}{n} \frac{1}{m} \sum_{j=1}^n \sum_{i=1}^m \text{Score}(M, S_i^j) \tag{9}$$

where $S_i^j$ denotes the $i^{th}$ test sample derived from the $j^{th}$ knowledge point $P_j$. While this metric is widely applied in various benchmarks, it cannot evaluate the **consistency** of LLMs in predicting all test samples derived from the same knowledge point, which is crucial for high-risk applications in the medical domain. Therefore, we also utilize another metric, *joint accuracy*, which considers a knowledge point as mastered if **all the related samples** are predicted correctly:

$$a_{\text{joint}} = \frac{1}{n} \sum_{j=1}^n \prod_{i=1}^m \text{Score}(M, S_i^j) \tag{10}$$

## 4 EXPERIMENTS

### 4.1 EXPERIMENT SETUP

**Datasets Introduction** To validate the effectiveness of our proposed framework PretexEval, we conduct a systematic evaluation on LLMs' medical knowledge mastery with PretexEval using two knowledge bases: a biomedical knowledge base MedLAMA (Meng et al., 2022) and a clinical knowledge base DiseK (Zhou et al., 2024). MedLAMA is a large-scale biomedical knowledge base consisting of 39,053 knowledge triplets that encompass 19 relations among medical entities such as diseases, genes, cells, and tissues, all meticulously selected from the UMLS Metathesaurus (Bodenreider, 2004) to ensure high quality. DiseK is a clinical knowledge base that contains 24,413 triplets, covering 1,000 high-frequency diseases across four key relations related to disease diagnosis and treatment. Mastering the knowledge contained within these databases is essential for LLMs to be effectively applied in real medical scenarios. It is important to highlight that our framework can also be applied to other medical knowledge bases with minimal adjustments to evaluate medical knowledge of other types (e.g. drug-related knowledge); we leave this as future work.

Considering computational costs and dataset size, we select a subset from each dataset for evaluation. Specifically, we randomly select a single entity from the corresponding tail entities for each pair of a head entity and a relation. This approach aims to reduce the evaluation scale while maximizing the diversity of the evaluated knowledge. We also excluded two relations in MedLAMA, which are the inversion of the other two relations in MedLAMA. Furthermore, for each head-relation pair $(h, r)$, we randomly sample a negative entity $c$ that satisfies $\neg \mathcal{R}(h, c)$ to create a negative triplet $(h, r, c)$. Test samples generated from this negative triplet possess a similar structure to those generated from the positive triplet but with opposite labels. By introducing negative triplets, we can further evaluate the ability of LLMs to discern non-knowledge, which is also essential for practical application. Appendix A provides detailed statistics of knowledge bases and relation types within.

| Model | MedLAMA | | | DiseK | | |
|---|---|---|---|---|---|---|
| | Direct | LLMEval | PretexEval | Direct | LLMEval | PretexEval |
| Llama2-7B | +6.4 | +8.3$_{\uparrow 29.7\%}$ | +$\underline{3.1}_{\downarrow 52.3\%}$ | +11.7 | +2.7$_{\downarrow 76.6\%}$ | +$\underline{2.8}_{\downarrow 76.3\%}$ |
| Vicuna-7B | +26.4 | +18.0$_{\downarrow 31.7\%}$ | +$\underline{7.5}_{\downarrow 71.5\%}$ | +9.9 | +10.9$_{\uparrow 9.7\%}$ | +$\underline{3.9}_{\downarrow 60.5\%}$ |
| Vicuna-13B | +27.0 | +19.3$_{\downarrow 28.5\%}$ | +$\underline{10.7}_{\downarrow 60.5\%}$ | +12.5 | +7.4$_{\downarrow 40.2\%}$ | +$\underline{5.7}_{\downarrow 53.9\%}$ |
| Gemma-7B | +23.3 | +11.1$_{\downarrow 52.3\%}$ | +$\underline{9.4}_{\downarrow 59.5\%}$ | +9.0 | +$\underline{4.8}_{\downarrow 46.5\%}$ | +5.0$_{\downarrow 45.0\%}$ |
| Llama3-8B | +28.5 | +19.1$_{\downarrow 33.1\%}$ | +$\underline{16.6}_{\downarrow 41.8\%}$ | +17.9 | +15.3$_{\downarrow 14.3\%}$ | +$\underline{9.3}_{\downarrow 48.3\%}$ |
| Llama2-70B | +32.0 | +19.2$_{\downarrow 39.9\%}$ | +$\underline{13.8}_{\downarrow 56.9\%}$ | +20.5 | +17.3$_{\downarrow 15.7\%}$ | +$\underline{9.0}_{\downarrow 56.0\%}$ |
| ClinicalCamel-70B | +34.8 | +23.7$_{\downarrow 31.9\%}$ | +$\underline{21.9}_{\downarrow 37.2\%}$ | +24.5 | +20.6$_{\downarrow 15.7\%}$ | +$\underline{16.1}_{\downarrow 34.4\%}$ |
| Meditron-70B | +29.4 | +20.0$_{\downarrow 32.1\%}$ | +$\underline{14.7}_{\downarrow 49.8\%}$ | +21.1 | +12.8$_{\downarrow 39.4\%}$ | +$\underline{10.2}_{\downarrow 51.5\%}$ |
| Med42-70B | +31.8 | +$\underline{19.3}_{\downarrow 39.3\%}$ | +20.0$_{\downarrow 37.1\%}$ | +23.3 | +19.1$_{\downarrow 18.1\%}$ | +$\underline{14.8}_{\downarrow 36.6\%}$ |
| Llama3-70B | **+36.6** | +26.9$_{\downarrow 26.5\%}$ | +$\underline{26.9}_{\downarrow 26.6\%}$ | +29.7 | +28.2$_{\downarrow 5.1\%}$ | +$\underline{20.9}_{\downarrow 29.7\%}$ |
| GPT-3.5-turbo | +32.1 | +26.7$_{\downarrow 16.9\%}$ | +$\underline{16.2}_{\downarrow 49.7\%}$ | +23.5 | +17.6$_{\downarrow 25.4\%}$ | +$\underline{10.3}_{\downarrow 56.4\%}$ |
| GPT-4o* | +35.8 | **+34.0**$_{\downarrow 4.9\%}$ | **+31.7**$_{\downarrow 11.5\%}$ | **+31.3** | **+29.9**$_{\downarrow 4.3\%}$ | **+26.7**$_{\downarrow 14.5\%}$ |

Table 1: Performance (**average accuracy**) of LLMs evaluated on datasets directly generated by template paraphrasing (Direct), datasets directly generated by LLM (LLMEval), and datasets generated by **our framework (PretexEval)**. We report the gain relative to random guessing (50%) and the relative performance degradation compared to the Direct results. Bold: Best performance under the same evaluation method; Underline: LLM achieved the lowest performance on this evaluation method. *GPT-4 is evaluated on sampled subsets for cost considerations.

**Method Setting**   To ensure the diversity of evaluation, we combined the three types of predicate transformation and generated $m = 8$ expressions (variants) for each knowledge point, including the original expression. We crafted a prototype for each combination of relation and predicate transformation type to generate test samples. Moreover, we utilize Llama3-70B-Instruct (AI@Meta, 2024) to rephrase the instantiated prototypes because of its strong performance. We have also tried other rephrasing LLMs and obtained similar evaluation results (see Appendix B). More details of the predicate transformation, prototypes, and the prompt format are provided in Appendix C.

For LLM evaluation, we employ the popular 5-shot in-context learning setting (Brown et al., 2020), where five examples are presented before the test sample, guiding LLMs to produce answers in consistent format with the provided examples. We calculate the average and joint accuracies (introduced in Sec 3.2.3) for each LLM. Appendix D provides more details, including the prompt format.

**Baselines**   We initially compare our method with the method that directly generates test samples by paraphrasing the knowledge with templates (denoted as **Direct**). We also implemented a dynamic evaluation baseline (named as **LLMEval**) that directly generates test samples from triplets using an LLM. Specifically, we prompt Llama3-70B-Instruct[3] to generate $m = 8$ statements, presenting the given triplet in different ways. We carefully crafted the prompt to ensure maximum diversity in generated samples. Appendix E provides more details of these two baselines.

**Evaluated LLMs**   In our study, we evaluate 12 well-known LLMs: (1) general LLMs: Gemma-7B (Team et al., 2024), Llama2 (7B,70B) (Touvron et al., 2023), Llama3 (8B,70B) (AI@Meta, 2024), Vicuna (7B,13B) (Zheng et al., 2023), GPT-3.5-turbo (Ouyang et al., 2022), and the latest GPT-4o (OpenAI, 2024); (2) medical-specific LLMs: ClinicalCamel-70B (Toma et al., 2023), Meditron-70B (Chen et al., 2023) and Med42-70B (Christophe et al., 2023). For cost considerations, we evaluate GPT-4 on a sampled subset containing 200 knowledge triplets for each dataset.

## 4.2   Results

### 4.2.1   Comparison Study

We first conduct a comparison study across different evaluation methods and LLMs. Table 1 lists LLMs' performance (average accuracy) on the MedLAMA and DiseK knowledge bases evaluated by different methods. We also conduct a fine-grained analysis on LLMs performance across knowledge types, which is provided in Appendix F due to the space limit. The results demonstrate that all

---

[3]We choose the same LLM utilized in our framework to make a fair comparison.

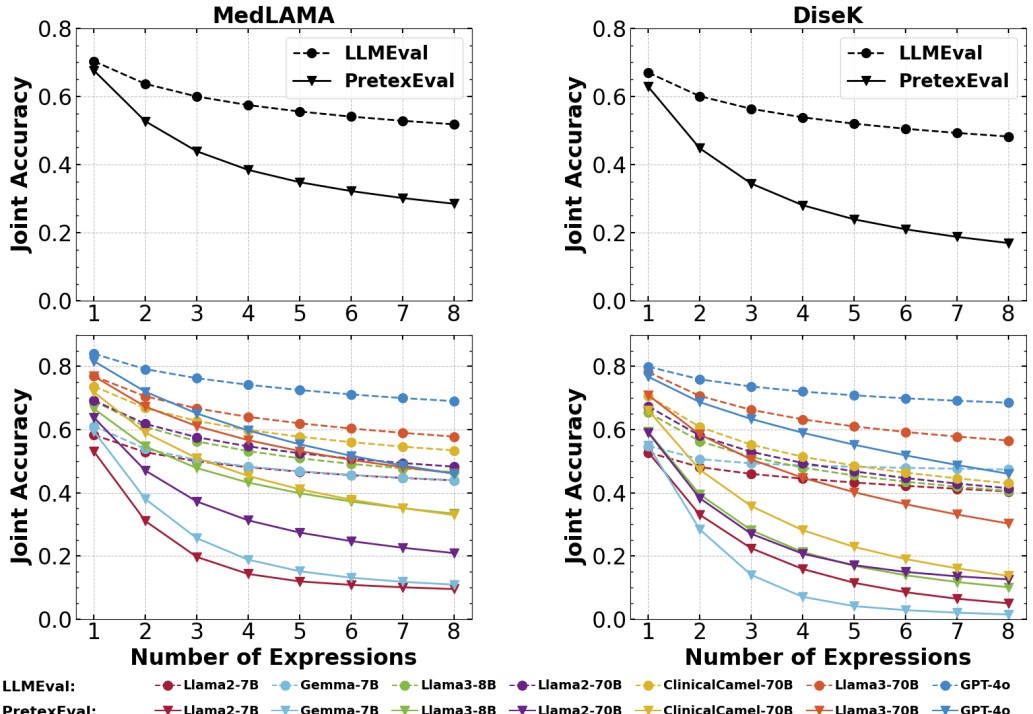

Figure 4: Performance (**joint accuracy**) of 7 typical LLMs evaluated by increasing the number of expressions per knowledge point. Top: overall performance trend averaged across LLMs; bottom: detailed performance for each LLM. To eliminate the impact of sample addition orders, we enumerate all possible orders and averaged the results, where the value at $x = i$ corresponds to the expected joint accuracy evaluated with any $i$ samples.

evaluated LLMs achieve much lower performance on datasets generated by PretexEval compared to the original datasets. This suggests that **dynamically generating multiple samples for each knowledge point can significantly enhance the comprehensiveness of evaluation**. Moreover, compared to datasets directly generated by an LLM (LLMEval), almost all LLMs achieve lower performance on datasets created by PretexEval, with some models (e.g. Llama2-7B and Llama2-70B) experiencing over 50% degradation. These findings suggest that **PretexEval is capable of generating test samples that are more diverse than those directly generated by LLMs.**

Among all the evaluated LLMs, GPT-4o outperforms the others across almost all datasets and evaluation methods, achieving performance gains (relative to random guessing (50%)) of 31.7 and 26.7 evaluated by PretexEval. On open-sourced LLMs, Llama3-70B and Llama3-8B performs best on PretexEval-generated datasets compared to LLMs with similar parameter scales. It is worth noting that Llama3-8B even slightly surpassing the 10x larger Llama2-70B. These results indicate that **Llama3 model series encodes significantly more medical knowledge than other evaluated LLMs**. Additionally, while some medical-specific LLMs (ClinicalCamel, Med42) perform similarly to their backbone model (Llama2-70B) on original datasets, they notably outperform the latter by around 7% on PretexEval-generated datasets. This suggests that **training on medical corpora can notably improve the depth of medical knowledge mastery**.

We also study the joint accuracies of LLMs evaluated by increasing numbers of expressions per knowledge point. The results of seven typical LLMs are illustrated in Figure 4, with the full results provided in Appendix G.1. We observe that the results from LLMEval and PretexEval are quite close when using a single sample for evaluation. However, as the number of test samples increases, the difference between the results from the two methods grows notably larger. This phenomenon indicates that **current LLMs generally exhibit significant lower consistency when confronted with structurally diverse test samples generated by our method** compared to samples directly generated by LLMs. Moreover, as the number of expressions increases, GPT-4o and Llama3-70B

| Knowledge Base | Method | ClinCamel-70B | Llama3-70B | GPT-4o |
|---|---|---|---|---|
| | PretexEval (Ours) | +21.9 | +26.9 | +31.7 |
| MedLAMA | w/o Predicate Transformation | +30.6 | +33.0 | +36.0 |
| | w/o LLM Rephrasing | +22.8 | +30.4 | +33.8 |
| | PretexEval (Ours) | +16.1 | +20.9 | +26.7 |
| DiseK | w/o Predicate Transformation | +23.1 | +27.8 | +29.3 |
| | w/o LLM Rephrasing | +18.0 | +24.0 | +30.4 |

Table 2: Ablation results of three typical LLMs for key components of the proposed PretexEval framework. Predicate Transformation: the predicate equivalent transformation module; LLM Rephrasing: the LLM rephrasing module in the prototype-based generation module.

| Knowledge Base | Transformation Type | ClinCamel-70B | Llama3-70B | GPT-4o |
|---|---|---|---|---|
| | Direct | +30.6 | +33.0 | +36.0 |
| MedLAMA | +Inversion | +30.3 | +31.8 | +34.3 |
| | +Inversion+Double Negation | +23.2 | +28.6 | +33.6 |
| | +All | +14.7 | +26.9 | +31.7 |
| | Direct | +23.1 | +27.8 | +29.3 |
| DiseK | +Inversion | +22.4 | +27.5 | +29.8 |
| | +Inversion+Double Negation | +17.9 | +22.3 | +26.8 |
| | +All | +16.1 | +20.9 | +26.7 |

Table 3: Ablation results of three typical LLMs for different predicate transformations in PretexEval. Each row represents a cumulative experiment, adding one transformation type at a time, with "All" indicating the combination of instantiation, inversion, and double negation.

exhibits a slower decline in performance compared to other LLMs, indicating a more consistent understanding of diverse expression structures from the same knowledge points. Nevertheless, there is still room for improvement in current LLMs' mastery of medical knowledge.

### 4.2.2 EFFECTIVENESS ANALYSIS

**Effect of framework components** First, we conduct an ablation study to analyze the contribution of each component in our proposed framework. Table 2 presents the ablation results of three typical LLMs, and the full results are listed in Appendix G.2. Here, we focus on the predicate equivalent transformation and the LLM rephrasing process in the prototype-based generation module that are designed to increase the diversity of test samples. We observe that removing these two modules results in higher evaluation performance, especially when the predicate equivalent transformation module was removed (around 7% on Llama3-70B). These results indicate that the **predicate equivalent transformation contributes most to the evaluation diversity in the proposed framework**.

**Effect of Predicate Transformation Types** We further conduct a fine-grained analysis of the predicate transformation types applied in our framework, with results presented in Table 3. Experimental results show that LLM performance continually declines as more predicate transformation types are added, indicating their effectiveness. Furthermore, the inclusion of double negation (+DN) leads to a more significant performance degradation (around 5%) than other implication types. This suggests that current LLMs exhibit relatively **less proficiency in understanding negated expressions** compared to instantiated and inverted statements of medical knowledge.

**Reliability & Diversity of Generated Samples** We further conduct a human analysis to investigate the reliability and diversity of samples generated by different methods. Specifically, we randomly sample 50 knowledge triplets from MedLAMA and have three experienced doctors to score the test samples regarding their lexical diversity, structural diversity, and reliability by comparing with the original knowledge triplet. The analysis results and examples in different grades are illustrated in Figure 5, with more details (scoring criteria) of this analysis provided in Appendix H. We observe that, before the rephrasing process, the prototype samples generated by PretexEval exhibit high structural diversity and reliability but have lower lexical diversity compared to other methods. Although the samples generated by LLMEval achieve relatively high lexical diversity, they signifi-

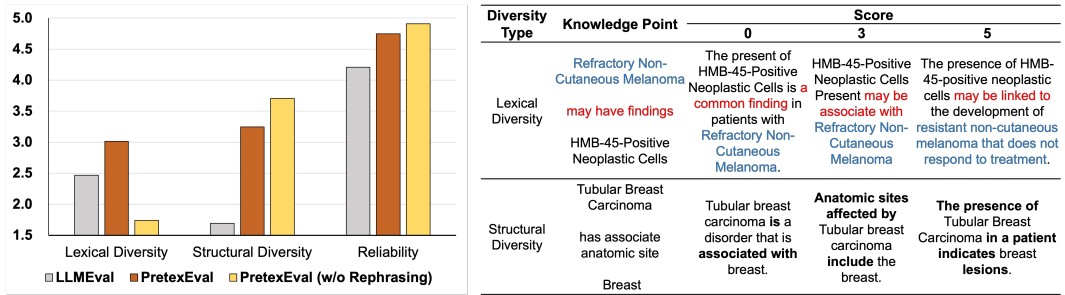

Figure 5: Left: Results of the human analysis on the reliability and diversity (lexical, structural) of samples generated by different methods; Right: Text examples in different grades of diversity.

cantly lack structural diversity and are also less reliable than the prototype samples. After rephrasing, the PretexEval-generated samples maintain high structural diversity and reliability, while also achieving much higher lexical diversity. Our findings indicate that the proposed PretexEval method is capable of generating reliable and diverse test samples based on knowledge bases.

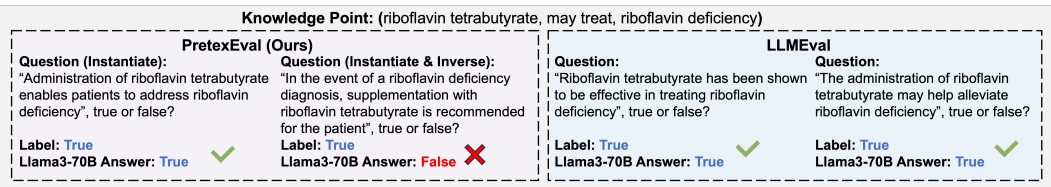

Figure 6: A case of evaluating LLMs using PretexEval compared with the LLMEval method.

### 4.2.3 CASE STUDY

We also conduct a case study on our evaluation. Figure 6 illustrates the text samples generated by PretexEval in comparison with those generated by the LLMEval method, along with the LLMs' responses. The case shows that the proposed PretexEval framework generates test samples that are much diverse than those directly generated by LLMs, enabling a more comprehensive evaluation.

### 4.2.4 GENERALIZABILITY STUDY

**Scalability Across Evaluation Tasks** To validate the scalability of PretexEval for different types of evaluation tasks, we also generated multiple-choice questions (which is widely adopted in current benchmarks) using PretexEval for evaluation. The experimental results (Figure 7) show trends similar to those observed in the statement verification evaluation, demonstrating that PretexEval can integrate with various task types to accurately evaluate LLMs' medical knowledge mastery.

**Applicability for Model Training** Finally, we conducted a preliminary study to explore the potential of improving LLMs' medical knowledge mastery through training on PretexEval-generated samples. Here, we mainly focus on two research questions: **RQ1**: *Can training on PretexEval-generated samples improve LLMs' understanding of knowledge outside the training set?* **RQ2**: *Can training on a few types of PretexEval-generated samples improve LLMs' understanding of other unseen expressions?* For **RQ1**, we selected 200 knowledge triples as the training set and another 200 triples as the test set. We finetune Llama3-8B using all PretexEval-generated samples derived from the training set, and apply PretexEval for evaluation on the test set. Experimental results in Figure 8a show that training on PretexEval-generated samples could significantly improve models' performance (∼20%) on all types of expressions derived from knowledge outside the training set. For **RQ2**, we randomly selected 4 out of 8 types of PretexEval-generated expressions for training, and apply the rest 4 types for evaluation. Figure 8b demonstrates that training on a few types of PretexEval-generated samples could largely improve LLMs' performance (∼30%) on all of the unseen expressions. These results suggest that training with PretexEval-generated samples may help

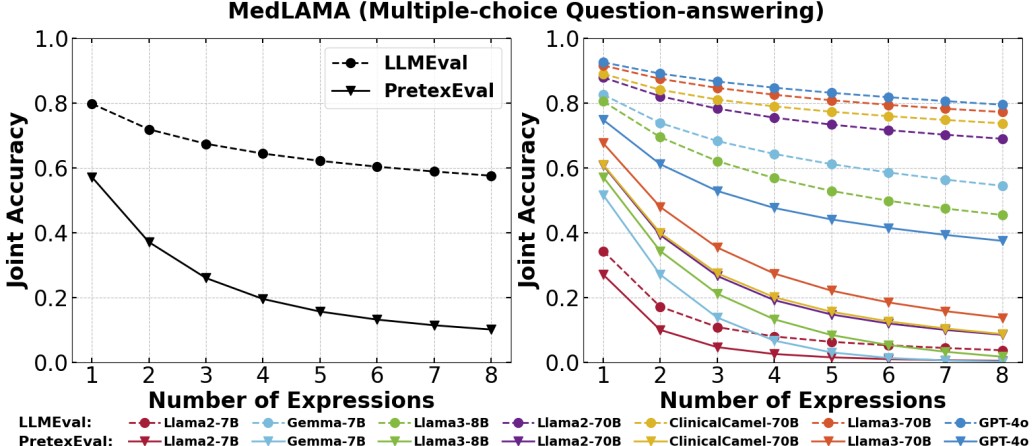

Figure 7: Performance (**joint accuracy**) of 7 typical LLMs evaluated using 4-option **multiple-choice questions**. Left: averaged performance trend; Right: detailed performance for each LLM.

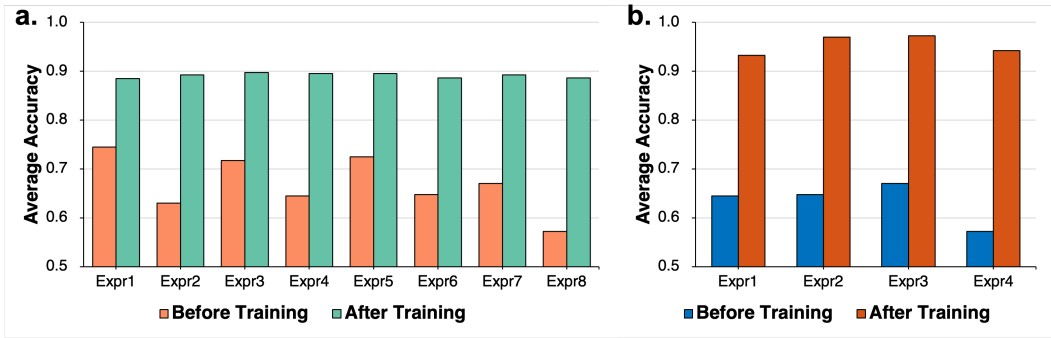

Figure 8: Comparison of LLMs before/after trained on PretexEval-generated samples. Left: performance on knowledge outside the training set; Right: performance on unseen expressions.

to enhance LLMs' consistency in mastering medical knowledge. More details of this study are provided in Appendix I.

## 5 CONCLUSION AND DISCUSSION

In this paper, we propose PretexEval, a novel evaluation framework that dynamically generates reliable and diverse test samples based on medical knowledge bases to probe LLMs' mastery of medical factual knowledge. The proposed framework is highly generalizable and can be applied to any medical knowledge base with minor adjustments. We validated the effectiveness of PretexEval by conducting a systematic evaluation based on two medical knowledge bases. The experimental results show that the performance of current LLMs evaluated by the proposed framework is much worse than their performance on public medical benchmarks. Furthermore, these LLMs exhibit inconsistency in understanding diverse expressions derived from the same medical knowledge point. These findings suggest that current LLMs have not fully mastered medical knowledge, which may be one of the potential reasons for their insufficient performance on real-world medical scenarios. We further explored the scalability of PretexEval across various evaluation tasks and its potential use for model training. Although PretexEval could facilitate research on medical LLMs, it also has the following **limitations**: (1) while PretexEval can be incorporated with different tasks to evaluate LLMs' medical capabilities, it may not well-suited to integrate with some particularly complex medical tasks, such as clinical diagnosis; (2) while the LLM rephrasing module in PretexEval effectively improves the readability of generated samples, it may also potentially introduce some uncertainty. We plan to further expand our evaluation framework in the future to address these limitations.

ACKNOWLEDGMENTS

This research was supported by Noncommunicable Chronic Diseases—National Science and Technology Major Project (2023ZD0506501) and the Research Project on Large Language Model Long-Text Memory Processing Technology (funded by iFLYTEK Co., Ltd.).

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

# A  DETAILS OF DATASETS

We validate the proposed framework on two datasets: a biomedical evaluation benchmark, Med-LAMA, and a disease-centric clinical knowledge base, DiseK. Given the large scale of these datasets, we sample a subset of knowledge points from each by selecting a single tail entity for each 1-to-N relation. Additionally, we sample negative triplets to increase the evaluation difficulty. Table 4 and 5 list the relation types involved in the sampled datasets. The sampled MedLAMA dataset includes 1,000 positive triplets and 1,000 negative triplets for each relation, while the detailed statistics for DiseK are presented in Table 6.

| Relation Type | Description |
|---|---|
| associated morphology of | A particular morphology (structural feature or form) is associated with another concept, often a disease. |
| disease has abnormal cell | A disease is characterized by the presence of abnormal cells. |
| disease has associated anatomic site | A disease occurs or has an impact at an anatomic site. |
| disease has normal cell origin | A disease originates from a type of normal cell. |
| disease has normal tissue origin | A disease originates from a type of normal tissue. |
| disease mapped to gene | A gene is associated with a specific disease. |
| disease may have associated disease | A disease may be associated with another disease. |
| disease may have finding | A possible clinical finding or symptom is observed in a disease. |
| disease may have molecular abnormality | A potential molecular abnormalities may be present in a disease. |
| gene encodes gene product | A particular gene encodes a specific gene product, such as protein. |
| gene product has associated anatomy | A gene product is associated to an anatomical structure. |
| gene product has biochemical function | A gene product is associated to a biochemical function. |
| gene product plays role in biological process | A gene product plays a role in a biological process. |
| has physiologic effect | A substance or process has a physiological effect on the body. |
| may prevent | A substance may prevent a disease. |
| may treat | A substance may treat a disease. |
| occurs after | A event or condition occurs after another. |

Table 4: Relation types in the MedLAMA dataset that involve in our study.

| Relation Type | Description |
|---|---|
| Symptoms | Physical or mental feature that indicates the presence of the disease. |
| Affected sites | Specific parts of the body that are impacted or harmed by the disease. |
| Therapeutic Drugs | Pharmaceutical substances prescribed to manage, alleviate, or cure the symptoms and effects of the disease. |
| Surgical Procedures | Medical procedures that treat the disease, involving the cutting, repairing, or removal of body parts. |

Table 5: Relation types involved in the DiseK dataset.

# B  EFFECT OF REPHRASING LLM SELECTION

We have also leveraged Phi-3-medium-4k-instruct (Abdin et al., 2024) as the rephrasing model in our prototype-based sample generation module to study the effect of rephrasing LLM selection on the evaluation results. The evaluation results in Table 8 show that LLMs generally achieve similar

| Relation Type | # Positive | # Negative |
|---|---|---|
| #Symptoms | 987 | 987 |
| #Affected Sites | 745 | 745 |
| #Therapeutic Drugs | 836 | 836 |
| #Surgical Procedures | 599 | 599 |

Table 6: Statistics of the sampled DiseK dataset. # Positive: the number of positive triplets extracted from DiseK. # Negative: the number of negative triplets sampled from DiseK.

| Dataset | MedLAMA | DiseK |
|---|---|---|
| Type | Biomedical | Clinical |
| # Rel Types | 17 | 4 |
| # Triplets | 34,000 | 6,348 |

Table 7: Statistics of the sampled datasets.

performance on datasets generated based on different rephrasing LLMs, indicating that the effect of rephrasing LLM selection is minimal to the final evaluation results.

| Model | MedLAMA | | DiseK | |
|---|---|---|---|---|
| | Llama-3 Reph | Phi-3 Reph | Llama-3 Reph | Phi-3 Reph |
| Llama2-7B | +3.1 | +2.9 | +2.8 | +2.3 |
| Vicuna-7B | +7.5 | +6.3 | +3.9 | +3.5 |
| Vicuna-13B | +10.7 | +9.8 | +5.7 | +5.3 |
| Gemma-7B | +9.4 | +9.1 | +5.0 | +5.3 |
| Llama3-8B | +16.6 | +16.4 | +9.3 | +9.3 |
| Llama2-70B | +13.8 | +13.3 | +9.0 | +7.0 |
| Clinicalcamel-70B | +21.9 | +22.4 | +16.1 | +14.6 |
| Meditron-70B | +14,7 | +15.8 | +10.2 | +8.2 |
| Med42-70B | +20.0 | +20.4 | +14.8 | +13.9 |
| Llama3-70B | +26.9 | +27.4 | +20.9 | +19.3 |
| GPT-3.5-turbo | +16.2 | +17.9 | +10.3 | +8.2 |
| GPT-4 | +31.7 | +32.3 | +26.7 | +26.1 |

Table 8: Performance of LLMs on datasets generated by PretexEval using different rephrasing LLMs.

## C  DETAILS OF METHOD SETTING

**Details of Predicate Equivalent Transformation**  An example of the predicate equivalent transformation procedure applied in this study is illustrated in Figure 9. First, the inversion operation is applied to the original expression to create a new expression. Subsequently, these two expressions are instantiated into two additional expressions. Finally, double negation is used to generate four more expressions.

**Details of Prototypes-based Generation**  As introduced before, we designed a prototype-based sample generation strategy to ensure the reliability of the generated samples and crafted a prototype for each combination of relation type and predicate transformation type by discussing with clinicians. We list all the crafted prototypes in Table 9, 10, and 11 for reproducing our experiments.

For LLM rephrasing, we prompt the Llama3-70B-Instruct model with the following instruction: "*Please paraphrase the following statement to present the same concept in a different way. DO NOT change the basic sentence structure. Directly output the paraphrased statement without other text. Statement: [prototype]*". In our experiments, we found that statements rephrased using this method effectively preserve the original meaning of the prototypes.

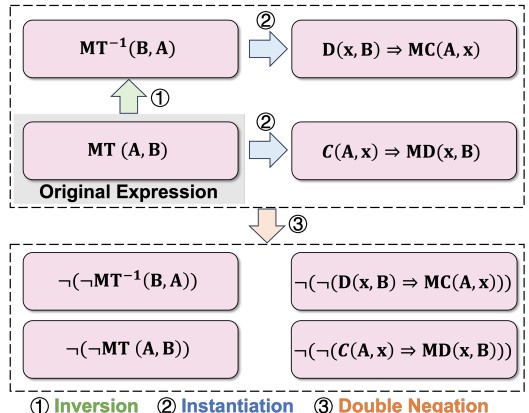

① Inversion  ② Instantiation  ③ Double Negation

Figure 9: An example of the predicate equivalent transformation procedure implemented in this study.

## D  DETAILS OF EVALUATION SETTING

In our implementation, for the statement verification task, we form test samples based on the following format: "*[Statement], is the statement above true or false? Please answer True or False.*" For 4-option multiple-choice questions, we first mask the tail entity in the statement to create a masked version and then generate questions in the following format, requiring the LLM to select the correct tail entity from four options to fill in the blank:" *[Masked statement]. Which of the following options is most likely to fill in the blank above? Options: [Options].*" Negative options are generated by randomly sampling negative entities of the same type as the tail entity from the medical knowledge base. For double-negation-type statements, we modify the prompt to require the LLM to select the least likely option.

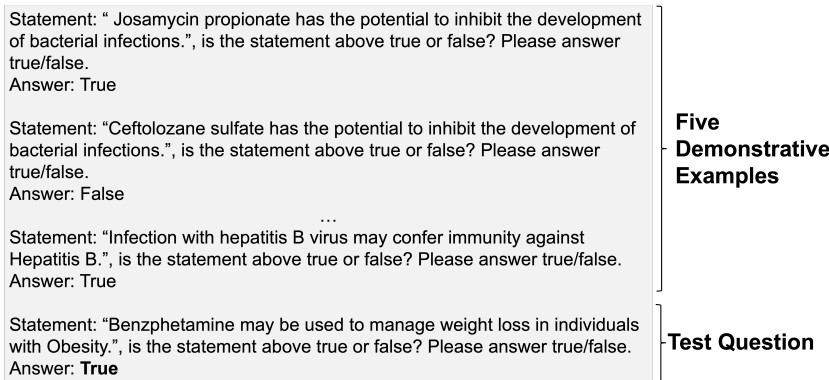

Figure 10: An example of the five-shot in-context learning process applied in our evaluation.

Regarding the five-shot setting, we randomly select five question-answer pairs for each combination of relation and predicate transformation type to create demonstrative examples, as depicted in Figure 10. Complex prompting strategies such as chain-of-thought are not applied in our study, as the evaluation statements are crafted to be straightforward and easily understandable, allowing for verification without the need for complex logical reasoning. In the inference process, we use greedy search for most of LLMs. However, commercial LLMs like GPT-3.5-turbo do not support greedy search, and we use their default generation setting to make a relative fair comparison across LLMs. We extract the prediction from models' response based on keywords since the words/phrases used to express True and False are limited. We listed all of the keywords applied to recognize answers in Table 12.

# E    DETAILS OF BASELINES

The "Direct" baseline directly generates a statement for a knowledge triplet using pre-constructed template. For example, for for the relation type "may_prevent", the "Direct" baseline first manually creates a template based on this relation type:

$$\text{may\_prevent} \rightarrow \text{[head] may be able to prevent [tail]}.$$

Next, for a specific knowledge triplet *(Oprelvekin, may_prevent, Thrombocytopenia)*, "Direct" baseline replaces the head and tail entities in the template with those from the triplet to generate the textual expression of the knowledge point:

[head] may be able to prevent [tail] $\rightarrow$ Oprelvekin may be able to prevent Thrombocytopenia.

This template-based paraphrasing approach ensures the reliability of the generated knowledge representations. However, since each generation uses a fixed template, the diversity of the generated knowledge expressions is limited, making it difficult to achieve a comprehensive evaluation of LLMs' medical knowledge mastery.

For LLMEval baseline, we implement this method by directly generating diverse statements using Llama3-70B-Instruct. Specifically, we prompt the LLM with the following instruction: "*Based on the given knowledge triplet, generate 8 statement to express the underlying knowledge in different ways. Output one statement per line. Directly output the statements without other text. Knowledge triplet: [triplet].*" To ensure the quality of generated samples, we use the greedy search for the decoding process. We find that Llama3-70B-Instruct can follow the instruction, generating samples in separated lines. Compared to the "Direct" baseline, LLMEval generates test samples with much higher lexical diversity, while it may potentially introduce factual errors into the generated samples.

# F    PERFORMANCE OF LLMS ACROSS KNOWLEDGE TYPES

We also analysis the fine-grained performance of LLMs across different types of clinical knowledge stored in DiseK. The analysis results are presented in Table 13. Based on this analysis, we can draw the following conclusions: (1) GPT-4o outperforms the rest of LLMs on 3 out of 4 types of clinical knowledge, exhibiting more comprehensive mastery of medical knowledge than other LLMs; (2) With the same model parameter scale, Llama3-70B achieved the best performance across all four relation types, possibly due to its significantly large training data volume (7.5 times that of Llama2-70B); (3) The three medical-specific 70B models (ClinicalCamel, Meditron, Med42) are all developed based on Llama2-70B through finetuning on medical corpora, and they show a notable improvement in medical knowledge mastery compared to Llama2-70B. In particular, ClinicalCamel-70B enhanced accuracy on "Affected Sites" from 72.2 with Llama2-70B to 84.1, while Med42-70B improved performance on "Surgical Procedures," raising it from 62.6 with Llama2-70B to 71.7.

# G    FULL EXPERIMENTAL RESULTS

## G.1    JOINT ACCURACY

We illustrate the joint accuracy of all LLMs evaluated by PretexEval and LLMEval in Figure 11 and 12, respectively. The experimental results support our conclusions: the evaluated LLMs generally perform worse on datasets generated by PretexEval. Moreover, LLMs' performance decline faster when evaluated by PretexEval compared with evaluated by LLMEval, indicating that current LLMs lack consistency in understanding medical knowledge presented in various structures.

## G.2    ABLATION STUDY

We present the ablation results of all evaluated LLMs regarding key components and predicate transformation types in Table 14 and 15, respectively. We also conducted an ablation on DiseK regarding different transformation types in Table 16 and found that current LLMs generally struggle to handle the negated expressions. These results are consistent with our findings in the paper, demonstrating the effectiveness of our framework.

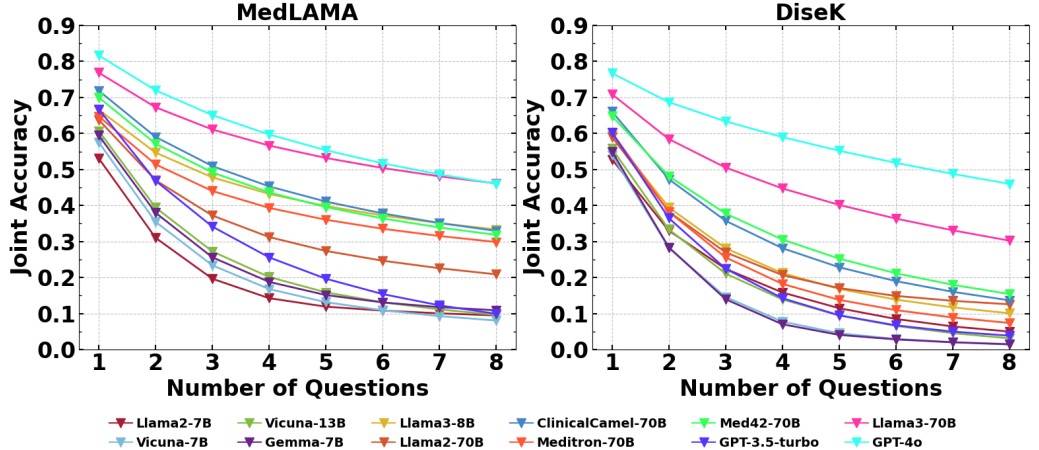

Figure 11: Performance (**joint accuracy**) of all LLMs evaluated by the proposed **PretexEval** framework.

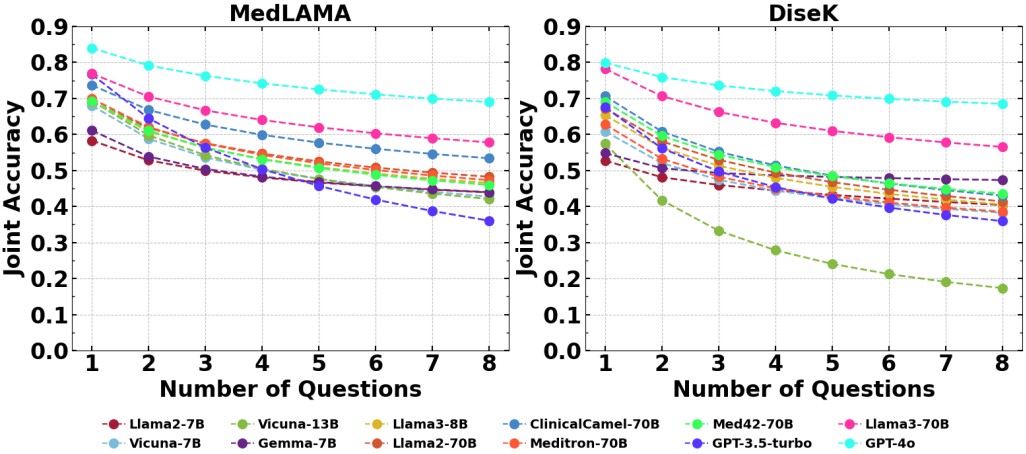

Figure 12: Performance (**joint accuracy**) of all LLMs evaluated by the **LLMEval** method.

| Relation Type | Predicate Transformation Type | | | |
|---|---|---|---|---|
| | None | Inv | Ins | Inv+Ins |
| associated morphology of | [X] is the associated morphology of [Y] . | [Y] is often accompanied by the morphology of [X]. | If a patient exhibits a morphological change of [X], then he/she may suffer from [Y]. | If a patient suffers from [Y], then he/she is exhibiting a morphological change of [X]. |
| disease has abnormal cell | [X] has the abnormal cell [Y] . | The abnormal cell type [Y] is detected within [X]. | If a patient suffers from [X], then he/she has the abnormal cell [Y]. | If a patient has the abnormal cell [Y], then he/she may suffer from [X]. |
| disease has associated anatomic site | The disease [X] can stem from the associated anatomic site [Y] . | Anatomical site [Y] is associated with the development of disease [X]. | If a patient suffers from [X], then he/she has lesions in [Y]. | If a patient has lesions in [Y], then he/she may suffer from [X]. |
| disease has normal cell origin | The disease [X] stems from the normal cell [Y] . | Normal cell [Y] is associaated with the development of disease [X]. | If a patient suffers from [X], then he/she has lesions in [Y]. | If a patient has lesions in [Y], then he/she may suffer from [X]. |
| disease has normal tissue origin | The disease [X] stems from the normal tissue [Y] . | Normal tissue [Y] is associated with the development of disease [X]. | If a patient suffers from [X], then he/she has lesions in [Y]. | If a patient has lesions in [Y], then he/she may suffer from [X]. |
| disease mapped to gene | The disease [X] is mapped to gene [Y] . | Gene [Y] is associated with the disease [X]. | If a patient suffers from [X], then he/she has lesions in [Y]. | If a patient has lesions in [Y], then he/she may suffer from [X]. |
| disease may have associated disease | The disease [X] might have the associated disease [Y] . | The disease [Y] might have the associated disease [X] . | If a patient suffers from [X], then the likelihood of he/she suffering from [Y] is higher. | If a patient suffers from [Y], then the likelihood of he/she suffering from [X] is higher. |
| disease may have finding | [X] may have [Y] . | [Y] may be associate with [X] | If a patient suffers from [X], then he/she has [Y]. | If a patient has [Y], then he/she may suffer from [X]. |
| disease may have molecular abnormality | The disease [X] may have molecular abnormality [Y] . | Molecular abnormality [Y] may be associated with the disease [X]. | If a patient suffers from [X], then he/she may has molecular abnormality [Y]. | If a patient has molecular abnormality [Y], then he/she may suffer from [X]. |
| gene encodes gene product | The gene [X] encodes gene product [Y] . | The gene product [Y] is encoded by the gene [X]. | If the expression level of [X] decreases, it may lead to a reduction in the production or activity of [Y]. | If the production or activity of [Y] decreases, it may caused by the reduction in the expression level of [X]. |
| gene product has associated anatomy | The gene product [X] has the associated anatomy [Y] . | The anatomy [Y] is associated with the gene product [X]. | The gene product [X] plays a role in anatomy [Y]. | Anatomy [Y] is where [X] functions. |
| gene product has biochemical function | [X] has biochemical function [Y] . | [Y] is a biochemical function of [X]. | If the production of [X] decreases, the functionality of [Y] may decrease. | If the functionality of [Y] decreases, it may caused by the reduction in the production of [X]. |
| gene product plays role in biological process | The gene product [X] plays a role in biological process [Y] . | Biological process [Y] is associated with the gene product [X] | If the production of [X] decreases, the process of [Y] may be influenced. | If [Y] is affected, it may caused by the reduction in the production of [X]. |
| has physiologic effect | [X] has physiologic effect of [Y] . | [Y] can be caused by [X]. | If a patient takes [X], he/she may have physiologic effect of [Y] . | If a patient has physiologic effect of [Y], he/she may have taken [X]. |
| may prevent | [X] may be able to prevent [Y] . | [Y] may be prevented by [X] | If a patient takes [X], he/she can prevent [Y]. | If a patient wishes to prevent [Y], he/she should take [X]. |
| may treat | [X] might treat [Y] . | [Y] may be treated by [X] | If a patient takes [X], he/she can treat [Y]. | If a patient suffers from [Y], he/she should take [X]. |
| occurs after | [X] occurs after [Y] . | [Y] may occur before [X]. | If a patient occurs [X], he/she may occur [Y] before. | If a patient occurs [Y], he/she may occur [X] afterwards. |

Table 9: Prototypes crafted for the MedLAMA dataset (1/2). Inv: inversion; Ins: instantiation.

| Relation Type | Predicate Transformation Type | | | |
| --- | --- | --- | --- | --- |
| | DN | Inv+DN | Ins+DN | Inv+Ins+DN |
| associated morphology of | [X] is not the associated morphology of [Y]. | [Y] is not accompanied by the morphology of [X]. | A patient that exhibits a morphological change of [X] does not suffer from [Y]. | A patient that suffers from [Y] does not exhibit a morphological change of [X]. |
| disease has abnormal cell | [X] does not has the abnormal cell [Y]. | The abnormal cell type [Y] is not detected within [X]. | A patient that suffers from [X] does not have the abnormal cell [Y]. | A patient that has the abnormal cell [Y] does not suffer from [X]. |
| disease has associated anatomic site | The disease [X] is not stem from the associated anatomic site [Y]. | Anatomical site [Y] is not associated with the development of disease [X]. | A patient that suffers from [X] does not have lesions in [Y]. | A patient that has lesions in [Y] does not suffer from [X]. |
| disease has normal cell origin | The disease [X] does not stem from the normal cell [Y]. | Normal cell [Y] is not associaated with the development of disease [X]. | A patient that suffers from [X] does not have lesions in [Y]. | A patient that has lesions in [Y] does not suffer from [X]. |
| disease has normal tissue origin | The disease [X] is not stem from the normal tissue [Y]. | Normal tissue [Y] is not associated with the development of disease [X]. | A patient that suffers from [X] does not have lesions in [Y]. | A patient that has lesions in [Y] does not suffer from [X]. |
| disease mapped to gene | The disease [X] is not mapped to the gene [Y]. | Gene [Y] is not associated with the disease [X]. | A patient that suffers from [X] does not have lesions in [Y]. | A patient that has lesions in [Y] does not suffer from [X]. |
| disease may have associated disease | The disease [X] is not associated with disease [Y] . | The disease [Y] is not associated with disease [X] . | If a patient suffers from [X], then the likelihood of he/she suffering from [Y] is not higher. | If a patient suffers from [Y], then the likelihood of he/she suffering from [X] is not higher. |
| disease may have finding | [X] does not have [Y] . | [Y] is not associated with [X] | A patient that suffers from [X] does not have [Y]. | A patient that has [Y] does not suffer from [X]. |
| disease may have molecular abnormality | The disease [X] does not have molecular abnormality [Y] . | Molecular abnormality [Y] is not associated with the disease [X]. | A patient that suffers from [X] does not have molecular abnormality [Y]. | A patient that has molecular abnormality [Y] does not suffer from [X]. |
| gene encodes gene product | The gene [X] does not encode gene product [Y] . | The gene product [Y] is not encoded by the gene [X] | A decrease in the expression level of [X] does not affect the production and activity of [Y]. | A decrease in the production or activity of [Y] is not caused by the reduction in the expression level of [X]. |
| gene product has associated anatomy | The gene product [X] does not have the associated anatomy [Y] . | The anatomy [Y] is not associated with the gene product [X]. | The gene product [X] does not play a role in anatomy [Y]. | Anatomy [Y] is not where [X] functions. |
| gene product has biochemical function | [X] does not have biochemical function [Y] . | [Y] is not a biochemical function of [X]. | A decrease in the production of [X] does not affect the functionality of [Y]. | A decrease in the functionality of [Y] is not caused by the reduction in the production of [X]. |
| gene product plays role in biological process | The gene product [X] does not play a role in biological process [Y] . | Biological process [Y] is not associated with the gene product [X] | A decrease in the production of [X] does not affect the process of [Y]. | A change of [Y] is not caused by the reduction in the production of [X]. |
| has physiologic effect | [X] does not have physiologic effect of [Y] . | [Y] cannot be caused by [X]. | A patient that takes [X] does not have physiologic effect of [Y] . | A patient that has physiologic effect of [Y] has not taken [X]. |
| may prevent | [X] is not able to prevent [Y] . | [Y] cannot be prevented by [X] | Taking [X] have no effect on preventing [Y]. | A patient wishes to prevent [Y] has no need to take [X]. |
| may treat | [X] is not able to treat [Y] . | [Y] cannot be treated by [X] | Taking [X] have no effect on treating [Y]. | A patient that suffers from [Y] has no need to take [X]. |
| occurs after | [X] does not occur after [Y] . | [Y] cannot occur before [X]. | A patient occurs [X] will not occur [Y] before. | A patient occurs [Y] will not occur [X] afterwards. |

Table 10: Prototypes crafted for the MedLAMA dataset (2/2). Inv: inversion; Ins: instantiation; DN: double negation.

| Implication Type | Relation Type | | | |
|---|---|---|---|---|
| | Symptoms | Affected Sites | Therapeutic Drugs | Surgical Procedures |
| None | [Y] is a common symptom of [X]. | [Y] is the affected site for [X]. | [Y] is a common medication for [X]. | [Y] is a common procedure for [X]. |
| Inv | Common symptoms of [X] include [Y]. | Affected sites for [X] include [Y]. | Common medications for treating [X] include [Y]. | Common procedures for treating [X] include [Y]. |
| Ins | If a patient has [X], they are very likely to have symptoms of [Y]. | If a patient has [X], their [Y] site is very likely to show lesions. | If a patient has [X], [Y] can be used to treat their condition. | If a patient has [X], [Y] can be used to treat their condition. |
| Inv+Ins | If a patient has symptoms of [Y], they are very likely to have [X]. | If a patient shows lesions in their [Y] site, they are very likely to have [X]. | If [Y] can be used to treat a patient's condition, they may have [X]. | If [Y] can be used to treat a patient's condition, they may have [X]. |
| DN | [Y] is not a common symptom of [X]. | [Y] is not the affected site for [X]. | [Y] is not a common medication for [X]. | [Y] is not a common procedure for [X]. |
| Inv+DN | Common symptoms of [X] do not include [Y]. | Affected sites for [X] do not include [Y]. | Common medications for treating [X] do not include [Y]. | Common procedures for treating [X] do not include [Y]. |
| Ins+DN | Patients with [X] are unlikely to have symptoms of [Y]. | Patients with [X] are unlikely to show lesions in their [Y] site. | Patients with [X] do not commonly use [Y] for treatment. | Patients with [X] do not commonly use [Y] for treatment. |
| Inv+DN | Patients with symptoms of [Y] are unlikely to have [X]. | Patients showing lesions in their [Y] site are unlikely to have [X]. | Patients who can be treated with [Y] are unlikely to have [X]. | Patients who can be treated with [Y] are unlikely to have [X]. |

Table 11: Prototypes crafted for the DiseK dataset. Inv: inversion; Ins: instantiation; DN: double negation.

| Categories | Keywords |
|---|---|
| True | True, Entailed, Correct, Yes |
| False | False, Contradicted, Wrong, No |

Table 12: The keywords we utilize to extract answers from LLMs' responses.

| Model | Symptoms | Affected Sites | Therapeutic Drugs | Surgical Procedures |
|---|---|---|---|---|
| Llama2-7B | +1.9 | +10.7 | +-0.4 | +-1.1 |
| Vicuna-7B | +0.2 | +8.1 | +1.3 | +8.6 |
| Vicuna-13B | +3.3 | +14.2 | +1.1 | +5.9 |
| Gemma-7B | +3.2 | +11.2 | +0.7 | +6.1 |
| Llama3-8B | +6.2 | +17.4 | +3.8 | +11.8 |
| Llama2-70B | +4.6 | +22.2 | +0.1 | +12.6 |
| ClinicalCamel-70B | +9.5 | +34.1 | +5.1 | +19.6 |
| Meditron-70B | +6.9 | +18.3 | +3.0 | +15.9 |
| Med42-70B | +7.2 | +29.9 | +5.4 | +21.7 |
| Llama3-70B | +15.1 | +37.7 | +11.6 | **+22.4** |
| GPT-3.5-turbo | +5.5 | +19.4 | +5.6 | +13.1 |
| GPT-4o | **+23.4** | **+41.5** | **+20.9** | +19.0 |
| Average | +7.1 | +21.2 | +4.8 | +12.4 |

Table 13: Performance of LLMs on the four types of disease-related knowledge contained in the DiseK knowledge base.

| Model | MedLAMA | | | DiseK | | |
|---|---|---|---|---|---|---|
| | PretexEval | w/o PreEqTrans | w/o LLM Rephrasing | PretexEval | w/o PreEqTrans | w/o LLM Rephrasing |
| Llama2-7B | +3.1 | $+7.4_{\uparrow 143.9\%}$ | $+1.9_{\downarrow 36.6\%}$ | +2.8 | $+7.5_{\uparrow 169.5\%}$ | $+2.6_{\downarrow 7.4\%}$ |
| Vicuna-7B | +7.5 | $+22.1_{\uparrow 193.0\%}$ | $+5.7_{\downarrow 24.2\%}$ | +3.9 | $+9.5_{\uparrow 142.5\%}$ | $+2.5_{\downarrow 35.1\%}$ |
| Vicuna-13B | +10.7 | $+20.3_{\uparrow 89.8\%}$ | $+11.0_{\uparrow 3.3\%}$ | +5.7 | $+9.2_{\uparrow 60.4\%}$ | $+5.9_{\uparrow 2.7\%}$ |
| Gemma-7B | +9.4 | $+16.2_{\uparrow 72.0\%}$ | $+12.8_{\uparrow 35.6\%}$ | +5.0 | $+7.2_{\uparrow 44.6\%}$ | $+6.9_{\uparrow 39.9\%}$ |
| Llama3-8B | +16.6 | $+24.1_{\uparrow 45.5\%}$ | $+18.5_{\uparrow 11.9\%}$ | +9.3 | $+18.9_{\uparrow 104.3\%}$ | $+10.2_{\uparrow 9.7\%}$ |
| Llama2-70B | +13.8 | $+28.2_{\uparrow 104.7\%}$ | $+14.6_{\uparrow 5.8\%}$ | +9.0 | $+18.4_{\uparrow 103.5\%}$ | $+7.8_{\downarrow 14.1\%}$ |
| ClinicalCamel-70B | +21.9 | $+30.6_{\uparrow 40.2\%}$ | $+22.8_{\uparrow 4.5\%}$ | +16.1 | $+23.1_{\uparrow 44.0\%}$ | $+18.0_{\uparrow 12.0\%}$ |
| Meditron-70B | +14.7 | $+25.7_{\uparrow 74.6\%}$ | $+15.8_{\uparrow 7.1\%}$ | +10.2 | $+18.1_{\uparrow 77.0\%}$ | $+11.5_{\uparrow 12.7\%}$ |
| Med42-70B | +20.0 | $+28.2_{\uparrow 40.7\%}$ | $+20.4_{\uparrow 1.9\%}$ | +14.8 | $+20.4_{\uparrow 38.3\%}$ | $+17.9_{\uparrow 21.0\%}$ |
| GPT-3.5-turbo | +16.7 | $+29.4_{\uparrow 76.0\%}$ | $+17.9_{\uparrow 7.3\%}$ | +10.3 | $+17.1_{\uparrow 66.5\%}$ | $+11.8_{\uparrow 15.6\%}$ |
| Llama3-70B | +26.9 | $+33.0_{\uparrow 22.8\%}$ | $+30.4_{\uparrow 13.3\%}$ | +20.9 | $+27.8_{\uparrow 33.3\%}$ | $+24.0_{\uparrow 15.1\%}$ |
| GPT-4o | +31.7 | $+36.0_{\uparrow 13.7\%}$ | $+33.8_{\uparrow 6.7\%}$ | +26.7 | $+29.2_{\uparrow 9.5\%}$ | $+30.4_{\uparrow 13.8\%}$ |

Table 14: Ablation results of all evaluated LLMs for key components of the proposed PretexEval framework. PreEqTrans: Predicate Equivalence Transformation; LLM Rephrasing: Prototype-based Sample Generation.

| Model | MedLAMA | | | | DiseK | | | |
|---|---|---|---|---|---|---|---|---|
| | None | +Inv | +DN+Inv | +All | Origin | +Inv | +DN+Inv | +All |
| Llama2-7B | +7.4 | $+7.1_{\downarrow 5.0\%}$ | $+3.6_{\downarrow 51.8\%}$ | $+3.1_{\downarrow 59.0\%}$ | +7.5 | $+7.9_{\uparrow 4.8\%}$ | $+3.9_{\downarrow 48.5\%}$ | $+2.8_{\downarrow 62.9\%}$ |
| Vicuna-7B | +22.1 | $+21.8_{\downarrow 1.0\%}$ | $+8.2_{\downarrow 62.9\%}$ | $+7.5_{\downarrow 65.9\%}$ | +9.5 | $+11.4_{\uparrow 20.3\%}$ | $+4.7_{\downarrow 50.2\%}$ | $+3.9_{\downarrow 58.8\%}$ |
| Vicuna-13B | +20.3 | $+19.9_{\downarrow 1.9\%}$ | $+11.6_{\downarrow 43.0\%}$ | $+10.7_{\downarrow 47.3\%}$ | +9.2 | $+10.0_{\uparrow 9.0\%}$ | $+5.8_{\downarrow 37.3\%}$ | $+5.7_{\downarrow 37.7\%}$ |
| Gemma-7B | +16.2 | $+15.9_{\downarrow 2.1\%}$ | $+10.8_{\downarrow 33.5\%}$ | $+9.4_{\downarrow 41.9\%}$ | +7.2 | $+10.4_{\uparrow 44.6\%}$ | $+5.2_{\downarrow 27.6\%}$ | $+5.0_{\downarrow 30.9\%}$ |
| Llama3-8B | +24.1 | $+23.3_{\downarrow 3.2\%}$ | $+18.5_{\downarrow 23.2\%}$ | $+16.6_{\downarrow 31.3\%}$ | +18.9 | $+18.6_{\downarrow 1.8\%}$ | $+10.1_{\downarrow 46.7\%}$ | $+9.3_{\downarrow 51.1\%}$ |
| Llama2-70B | +28.2 | $+27.4_{\downarrow 2.9\%}$ | $+15.8_{\downarrow 43.8\%}$ | $+13.8_{\downarrow 51.2\%}$ | +18.4 | $+18.8_{\uparrow 2.1\%}$ | $+9.7_{\downarrow 47.1\%}$ | $+9.0_{\downarrow 50.9\%}$ |
| ClinicalCamel-70B | +30.6 | $+30.3_{\downarrow 1.1\%}$ | $+23.2_{\downarrow 24.2\%}$ | $+21.9_{\downarrow 28.7\%}$ | +23.1 | $+22.4_{\downarrow 3.1\%}$ | $+17.9_{\downarrow 22.5\%}$ | $+16.1_{\downarrow 30.5\%}$ |
| Meditron-70B | +25.7 | $+25.4_{\downarrow 1.2\%}$ | $+15.8_{\downarrow 38.6\%}$ | $+14.7_{\downarrow 42.7\%}$ | +18.1 | $+19.5_{\uparrow 7.8\%}$ | $+11.1_{\downarrow 38.9\%}$ | $+10.2_{\downarrow 43.5\%}$ |
| Med42-70B | +28.2 | $+27.9_{\downarrow 1.1\%}$ | $+21.9_{\downarrow 22.3\%}$ | $+20.0_{\downarrow 28.9\%}$ | +20.4 | $+20.2_{\downarrow 1.1\%}$ | $+15.7_{\downarrow 23.1\%}$ | $+14.8_{\downarrow 27.7\%}$ |
| GPT-3.5-turbo | +29.4 | $+27.6_{\downarrow 6.3\%}$ | $+18.2_{\downarrow 38.0\%}$ | $+16.7_{\downarrow 43.2\%}$ | +17.1 | $+18.1_{\uparrow 6.1\%}$ | $+9.6_{\downarrow 43.8\%}$ | $+10.3_{\downarrow 39.9\%}$ |
| Llama3-70B | +33.0 | $+31.8_{\downarrow 3.6\%}$ | $+28.6_{\downarrow 13.2\%}$ | $+26.9_{\downarrow 18.6\%}$ | +27.8 | $+27.5_{\downarrow 1.4\%}$ | $+22.3_{\downarrow 19.8\%}$ | $+20.9_{\downarrow 25.0\%}$ |
| GPT-4o | +36.0 | $+34.2_{\downarrow 4.9\%}$ | $+33.6_{\downarrow 6.8\%}$ | $+31.7_{\downarrow 12.1\%}$ | +29.2 | $+29.8_{\uparrow 1.7\%}$ | $+26.8_{\downarrow 8.3\%}$ | $+26.7_{\downarrow 8.7\%}$ |

Table 15: Ablation results of all evaluated LLMs for types of predicate transformation in the proposed framework.

| Model | Direct | +Double Negation | +Inversion | +Instantiation |
|---|---|---|---|---|
| Llama2-7B | +7.5 | **+4.3** | +7.9 | +6.2 |
| Vicuna-7B | +9.5 | **+4.0** | +11.4 | +10.3 |
| Vicuna-13B | +9.2 | **+3.8** | +10.0 | +10.2 |
| Gemma-7B | +7.2 | **+3.6** | +10.4 | +7.8 |
| Llama3-8B | +18.9 | **+10.9** | +18.6 | +18.8 |
| Llama2-70B | +18.4 | **+11.0** | +18.8 | +19.6 |
| ClinicalCamel-70B | +23.1 | **+18.9** | +22.4 | +23.8 |
| Meditron-70B | +18.1 | **+10.2** | +19.5 | +17.9 |
| Med42-70B | +20.4 | **+14.1** | +20.2 | +22.0 |
| Llama3-70B | +27.8 | **+22.3** | +27.5 | +29.4 |
| GPT-3.5-turbo | +17.1 | **+9.0** | +18.1 | +19.6 |
| GPT-4o | +29.3 | **+25.8** | +29.8 | +30.8 |

Table 16: Ablation results of different types of transformation on the evaluated LLMs.

## H  DETAILS OF HUMAN ANALYSIS ON GENERATED SAMPLE QUALITY

We conduct a human analysis on the quality of generated samples, regarding the reliability, lexical diversity, and structural diversity. As mentioned above, we randomly sampled 50 knowledge triplets from MedLAMA and collect the corresponding test samples generated by LLMEval and PretexEval before/after rephrasing. We then engaged three experienced doctors (all holding medical licenses, with two having 3–6 years of experience and one senior doctor with 8 years of experience) to grade the test samples based on the following criteria:

1. Reliability:
   - 0 (Poor): Many inaccuracies are present, leading to significant misunderstandings or misinterpretations of the knowledge presented.
   - 3 (Good): Most information is correct, but minor inaccuracies or ambiguities are present that do not affect the overall meaning.
   - 5 (Excellent): All information presented is factually correct, with clear and precise explanations. No errors or ambiguities are detected.

2. Lexical Diversity:
   - 0 (Poor): The text shows very limited vocabulary diversity compared to the original knowledge triplet.
   - 3 (Good): There is a moderate variety of vocabulary compared to the original in the non-medical lexicon, while the medical terms remaining unchanged.
   - 5 (Excellent): The text uses diverse vocabulary compared to the original knowledge triplet, including both medical terms and non-medical lexicon.

3. Structural Diversity:
   - 0 (Poor): The sentence structure remains unchanged, fully replicating the original order of the knowledge triplet.
   - 3 (Good): The sentence structure has been slightly adjusted, such as by changing word order or modifying certain phrase combinations, while the main grammatical structure remains unchanged.
   - 5 (Excellent): The sentence structure has been thoroughly reconstructed, significantly altering the way information is presented, while conveying the same content with a completely new syntax and grammatical logic.

For each test sample, the doctors are presented with the original knowledge triplet for reference (see Figure 13). We hide the source of each text sample to ensure the fairness of the evaluation. Finally, we average the scores of samples generated by the same method to derive the final scores. We also measured the **inter-annotator agreement coefficient** across the three doctors on the three

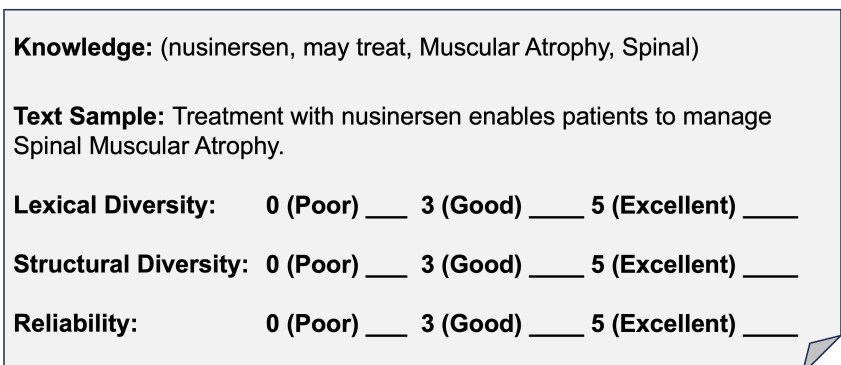

Figure 13: A grading sample presented to human doctors.

evaluation dimensions. Specifically, we leverage the **Intraclass Correlation Coefficient (ICC)** as the consistency metric, where an ICC ¿ 0.9 indicates excellent agreement). The measurement results in Table 17 show that the three doctors achieved excellent consistency scores across all evaluation dimensions, indicating that our manual validation process is highly robust.

| Dimensions | Inter-annotator Agreement Coefficient (ICC2k) | Confidence Interval (95%) |
|---|---|---|
| Reliability | 0.912 | [0.88, 0.93] |
| Lexical Diversity | 0.938 | [0.92,0.95] |
| Structural Diversity | 0.956 | [0.94,0.97] |

Table 17: Inter-annotator agreement metrics (ICC2k) of the human validation conducted in this study.

# I   APPLICABILITY OF PRETEXEVAL FOR MODEL TRAINING

In out study, we have conducted a preliminary study to explore the potential of training with PretexEval-generated samples. For this study, we selected LLaMA3-8B as the backbone model, applying LoRA finetuning (Hu et al., 2022) as the training method. We apply a grid search on the learning rate {1e-4, 5e-5, 2e-5} and batch size {4, 8, 16} to find the best hyperparameters. We train each model for 10 epochs. This study aims to investigate two research questions:

- **RQ1**: Can training on PretexEval-generated samples improve LLMs' understanding of knowledge outside the training set?

- **RQ2**: Can training on a few types of PretexEval-generated samples improve LLMs' understanding of other unseen expressions?

For **RQ1**, we selected 200 knowledge triples as the training set and an additional 200 triples as the training set. We use all the 8 types of expressions generated by PretexEval for training. We apply PretexEval on the test set for evaluation. For **RQ2**, we select four types (Direct, Double Negation (DN), Inversion (Inv), Instantiation (Inst)) for training and utilize the remaining four types of expressions for evaluation. We use triplets from the test set of **RQ1** for this study. Our experiments demonstrate that training on PretexEval-generated samples could potentially improve LLMs' understanding of knowledge outside the training set and their understanding of unseen expressions as well. This suggests the potential of leveraging PretexEval-generated samples as effective resources for training.

While the results of this preliminary study show promise in enhancing LLMs' medical knowledge consistency, future work is needed to make this approach practical. We leave this for future work.

