# OpenReview forum: "Reliable and Diverse Evaluation of LLM Medical Knowledge Mastery"
_ICLR.cc/2025/Conference — ICLR 2025 Poster_

### Official Review · Reviewer_D3So · 2024-11-02

**Soundness:** 3
**Presentation:** 3
**Contribution:** 3
**Rating:** 6
**Confidence:** 4

**Summary:**

The paper introduces PretexEval, a framework for generating test samples from medical knowledge bases by generating variants of predicate expressions. This is done in three steps:

(1) A knowledge point is converted to a predicate expression.

(2) A predicate expression is converted to multiple predicate variants.

(3) Predicate variants are converted back to a textual format.

Authors prompt a number of current LLMs to verify the correctness of each statement (True/False), and find that current LLMs display subpar performance on these newly generated test samples, leading to the conclusion that they have not yet mastered medical knowledge.

**Strengths:**

- The motivation of generating reliable test samples without factual errors, which is especially important in the medical domain, is clear and concise.
- The proposed framework can be applied to any medical knowledge base.

**Weaknesses:**

- Opposed to the overall impression that the proposed framework enables diverse evaluation of LLMs’ medical knowledge, the paper only utilizes a single metric (true/false verification) for evaluation. A reviewer think that addressing multiple choice question answering or generating through (subject, relation, object) triplet makes more beneficial to elicit LLMs' medical knowledge. Although the authors do involve joint accuracy as a measure to account for all related samples that stem from a specific knowledge point, defining the knowledge span of current LLMs without diverse evaluation metrics (e.g. Single/Multiple Choice QA, Extractive/Abstractive QA) seems insufficient.
- Further consideration on baseline selection seems necessary to support the efficacy of the proposed framework. Based on the initial problem statement that prompting LLMs to “directly generate test samples based on specific knowledge points” may lead to insufficient factuality, the LLMEval baseline from Table 1 seems to be unreliable. Could you provide more justification details why you added the LLMEval baseline?

**Questions:**

- In the Related Work section, authors point out that existing benchmarks crafted from medical experts may “become outdated or leaked to LLMs, affecting the comprehensiveness of evaluation”. How is the dataset created from the suggested framework any different? Further clarification regarding the authors’ intentions would be helpful.
- How are the test samples for the “Direct” baseline constructed? More clarification on the template paraphrasing process would be helpful, as readers may not be familiar with related works that adopt template paraphrasing. Please provide the example of a template and how it is used to create a test sample for the "Direct" baseline.
- Can you elaborate more on the qualifications of the “two experienced doctors” that have manually evaluated the test samples? Such as providing the years of experience or the doctors' specialities, or any relevant certifications that qualify them to evaluate these medical knowledge samples.
- When analyzing the “Effect of Predicate Transformation Types”, have the authors conducted the “Direct + Double Negation” setting as well? Experiments in this setting seems to be important for the claim that “current LLMs exhibit relatively less proficiency in understanding negated expressions”.

---

> ### Author Response · Authors · 2024-11-22
> **Response to Reviewer D3So (1/4)**
>
> We are sincerely grateful for your detailed and constructive feedback. Below, we provide our responses to each of the concerns you raised.
>
> 1. **Scalability across Evaluation Metrics**: Thank you for your insightful and constructive comments. In our work, we primarily use a true/false verification task as the evaluation metric, as it can be directly derived from the knowledge expressions generated by PretexEval. Nevertheless, it is worth noting that the proposed PretexEval framework is **highly scalable** in terms of evaluation metrics and can be incorporated with various task formats. To validate the scalability of PretexEval, we follow your constructive advice and conduct a preliminary study by further incorporating PretexEval with the multiple-choice question (MCQ) task. Specifically, we first randomly selected 200 knowledge triplets from one of the knowledge bases utilized in this study (MedLAMA knowledge base). Next, we generate 1,600 reliable and diverse samples using PretexEval, and further rephrase them into the corresponding MCQ questions. The experimental results (average accuracy gain compared to random guessing) are listed below:
>
>    | Models            | LLMEval | PretexEval (Ours) |
>    | ----------------- | :-----: | :---------------: |
>    | Llama2-7B         |  +9.4   |     **+2.1**      |
>    | Vicuna-7B         |  +25.7  |     **+12.1**     |
>    | Vicuna-13B        |  +36.3  |     **+12.5**     |
>    | Gemma-7B          |  +57.6  |     **+26.7**     |
>    | Llama3-8B         |  +55.5  |     **+32.2**     |
>    | Llama2-70B        |  +62.9  |     **+35.8**     |
>    | ClinicalCamel-70B |  +64.0  |     **+36.0**     |
>    | Meditron-70B      |  +62.6  |     **+30.2**     |
>    | Med42-70B         |  +66.3  |     **+39.1**     |
>    | Llama3-70B        |  +66.6  |     **+42.7**     |
>    | GPT-3.5-turbo     |  +38.6  |     **+16.7**     |
>    | GPT-4o            |  +67.6  |     **+49.8**     |
>
> We observe a strong correlation between LLMs' performance on MCQ questions and their performance on true/false verification tasks presented in our paper (Spearman rank correlation coefficient = 0.91, p-value = 4e-05). Moreover, the evaluated LLMs consistently achieve much lower performance on samples generated by PretexEval compared to those generated by LLMEval, reflecting a trend similar to the results reported in our paper. The experimental results above suggests that PretexEval can generate more diverse samples across different task settings to better evaluate LLMs' mastery of medical knowledge. The detailed results of this preliminary study is provided in Figure 7 of our revised paper.

---

> > ### Author Response · Authors · 2024-11-22
> > **Response to Reviewer D3So (2/4)**
> >
> > 2. **Justification of Baseline Selection**: We sincerely appreciate your thoughtful comments. Current evaluation benchmarks are typically in static forms, making LLM evaluations vulnerable to issues such as outdated evaluations and benchmark leakage. These issues have also received widespread attention in recent research. One effective strategy to address these problems is the dynamic generation of test samples. However, existing methods for dynamically generating test samples face issues such as insufficient diversity and reliability of the generated samples. Given the richness and reliability of medical knowledge contained in medical knowledge bases, which are regularly maintained by experts, this study primarily focuses on dynamically generating diverse and reliable samples based on medical knowledge bases. Here, LLMEval serves as a baseline method that leverages LLM prompting to directly generate test samples from medical knowledge. Specifically, we choose LLMEval as a baseline for the following reasons:
> >
> >    1. **Subjective Comparison**: The core objective of this study is to dynamically generate diverse and reliable evaluation samples based on a medical knowledge base, enabling a more comprehensive and in-depth evaluation of LLMs’ mastery of medical knowledge. A straightforward and intuitive approach is to directly use LLM prompting to generate evaluation samples based on medical knowledge points, as done in the LLMEval method. Through validation by medical experts, we compared LLMEval and our proposed PretexEval method in terms of reliability**, **lexical diversity, and structural diversity of the generated knowledge representations. The results revealed that LLMEval-generated samples had significantly lower structural diversity and reliability compared to PretexEval, demonstrating the effectiveness of PretexEval in dynamically producing more diverse and reliable evaluation samples.
> >
> >    2. **Objective Comparison**: While LLMEval exhibits insufficient reliability, this lack of reliability is relative to PretexEval, and our manual validation showed that LLMEval could still generate fundamentally correct knowledge expressions in ~90% of cases. Moreover, directly generating evaluation samples by prompting LLMs is also one of the common method in current research [1-2]. Therefore, comparing the evaluation results obtained using both PretexEval and LLMEval offers a more objective perspective to demonstrate the effectiveness of our framework. Our experimental results show that current LLMs generally performed worse on PretexEval-generated samples compared to LLMEval-generated ones, indicating that PretexEval generates more diverse and challenging test samples, thereby facilitating a better evaluation of LLMs’ medical knowledge mastery.
> >
> > [1] Li K, Zhang Y. Planning First, Question Second: An LLM-Guided Method for Controllable Question Generation. Findings of ACL 2024.
> >
> > [2] Li R, Li R, Wang B, et al. IQA-EVAL: Automatic Evaluation of Human-Model Interactive Question Answering. NeurIPS 2024.

---

> > > ### Author Response · Authors · 2024-11-22
> > > **Response to Reviewer D3So (3/4)**
> > >
> > > 3. **Clarification of Intention**: We greatly appreciate your detailed comments on the intentions behind our work. In this paper, we highlight two key issues that may arise with current static benchmarks crafted by medical experts: outdated knowledge and benchmark leakage.
> > >
> > >    1. **Outdated Knowledge**: Medicine is a constantly evolving field, with rapid advancements in new knowledge (such as the discovery of new pathogens, drugs, and vaccines) and the potential for existing knowledge to be overturned by new research. For example, hormone replacement therapy (HRT) was once widely considered a standard treatment for postmenopausal women to prevent cardiovascular disease and osteoporosis. However, after studies revealed that HRT could increase the risk of breast cancer, its clinical use significantly decreased. As a result, static medical benchmarks constructed by experts may become outdated over time and no longer accurately evaluate an LLM’s medical knowledge mastery. While expert re-annotations could theoretically update these benchmarks, the process is resource-intensive and time-consuming. Our proposed PretexEval framework, on the other hand, generates test samples based on knowledge points from medical knowledge bases, which are regularly maintained by experts and publicly accessible (e.g., UMLS, DrugBank). Therefore, our framework can effectively mitigate the problem of outdated knowledge and also dynamically generate diverse samples based on medical knowledge.
> > >
> > >
> > >    2. **Benchmark Leakage**: Benchmark leakage refers to the risk that publicly available benchmark data may be used by LLMs during training, causing the model to memorize the test samples and answers. As a result, these benchmarks may no longer provide reliable evaluations of LLMs’ capabilities. Several studies [1-3] have confirmed the existence of this leakage issue. Our method addresses this by employing predicate transformations to dynamically generate new test samples from the medical knowledge base. This ensures that LLMs cannot rely on memorizing specific test samples to answer questions, thus effectively preventing benchmark leakage.
> > >
> > > We greatly appreciate your valuable feedback. We have made updates to the Related Work section of the paper to further enhance the clarity of our intentions.
> > >
> > > [1] Magar I, Schwartz R. Data Contamination: From Memorization to Exploitation. ACL 2022.
> > >
> > > [2] Sainz O, Campos J, García-Ferrero I, et al. NLP Evaluation in trouble: On the Need to Measure LLM 	Data Contamination for each Benchmark. Findings of the EMNLP 2023.
> > >
> > > [3] Zhang W, Zhang R, Guo J, et al. Pretraining Data Detection for Large Language Models: A Divergence-based Calibration Method. EMNLP 2024.
> > >
> > > 4. **Clarification on the "Direct" baseline**: We are very grateful for your constructive suggestions. The “Direct” baseline first manually constructs a natural language template for each type of relationship in the knowledge base. Then, by filling in the head and tail entities into the template, it transforms medical knowledge triplets into evaluation samples. For example, for the relation type "may_prevent", the "Direct" baseline first manually creates a template based on this relation type:
> > >
> > > ```
> > > may_prevent -> [head] may be able to prevent [tail].
> > > ```
> > >
> > > Next, for a specific knowledge triplet *(Oprelvekin, may_prevent, Thrombocytopenia)*, "Direct" baseline replaces the head and tail entities in the template with those from the triplet to generate the textual expression of the knowledge point:
> > >
> > > ```
> > > [head] may be able to prevent [tail] -> Oprelvekin may be able to prevent Thrombocytopenia.
> > > ```
> > >
> > > This template-based paraphrasing approach ensures the reliability of the generated knowledge representations. However, since each generation uses a fixed template, the diversity of the generated knowledge expressions is limited, making it difficult to achieve a comprehensive evaluation of LLMs’ medical knowledge mastery. Thank you again for your constructive feedback. Due to the limitations of the paper’s length, we have provided additional details about this baseline in Appendix E to help readers better understand the specific process of template paraphrasing.

---

> ### Author Response · Authors · 2024-11-22
> **Response to Reviewer D3So (4/4)**
>
> 5. **Details of Doctor Qualification**: Thank you for your thorough review and suggestions. In this study, we initially employed two doctors to manually validate the reliability and diversity of the test samples. Both doctors hold medical licenses and have 3-6 years of clinical experience. During the discussion phase of the paper, we took Reviewer fVf4’s advice into account and further engaged a senior doctor with about 8 years of experience to evaluate the test samples. We also measured the **inter-annotator agreement coefficient** across the three doctors on the three evaluation dimensions (using the Intraclass Correlation Coefficient (ICC) as the consistency metric, where an ICC > 0.9 indicates excellent agreement). We provide the inter-annotator agreement coefficients and corresponding 95% confidence intervals in the table below. The results show that the three doctors achieved excellent consistency scores across all evaluation dimensions, indicating that our clinical expert validation process is highly robust. Due to page limitations, we have added information about the doctors’ qualifications to Appendix H of our paper.
>
>    | Dimensions           | Inter-annotator Agreement Coefficient | Confidence Interval |
>    | -------------------- | :-----------------------------------: | :-----------------: |
>    | Reliability          |                 0.912                 |    [0.88, 0.93]     |
>    | Lexical Diversity    |                 0.938                 |     [0.92,0.95]     |
>    | Structural Diversity |                 0.956                 |     [0.94,0.97]     |
>
> 6. **Details of Ablation Study**: We sincerely appreciate your insightful and constructive comments. In fact, we have previously analyzed LLMs’ performance on three different settings, each involving a specific transformation type: Direct+Double Negation, Direct+Inversion, and Direct+Instantiation. We observed that Double Negation generally leads to a larger performance reduction compared to the other two transformation types. For example, for GPT-4o, introducing Double Negation, Inversion, and Instantiation individually results in performance drops of 1.5%, 0.6%, and -0.6%, respectively. For LLaMA3-70B, the performance decreases by 4.0%, 0.8%, and -1.0% by introducing these three transformation types, respectively. Furthermore, we also conducted an ablation study on the DiseK knowledge base and listed the results below. We observed that current LLMs achieve much lower performance after introducing Double Negation compared to the other two transformation types, indicating that these LLMs generally exhibit relatively less proficiency in understanding negated expressions. Again, thank you for your valuable suggestions. Due to the page limitation, we have provided the detailed results of this ablation study in Appendix G of our revised paper.
>
>    | Model             | Direct | +Double Negation | +Inversion | +Instantiation |
>    | ----------------- | :----: | :--------------: | :--------: | :------------: |
>    | Llama2-7B         |  +7.5  |     **+4.3**     |    +7.9    |      +6.2      |
>    | Vicuna-7B         |  +9.5  |     **+4.0**     |   +11.4    |     +10.3      |
>    | Vicuna-13B        |  +9.2  |     **+3.8**     |   +10.0    |     +10.2      |
>    | Gemma-7B          |  +7.2  |     **+3.6**     |   +10.4    |      +7.8      |
>    | Llama3-8B         | +18.9  |    **+10.9**     |   +18.6    |     +18.8      |
>    | Llama2-70B        | +18.4  |    **+11.0**     |   +18.8    |     +19.6      |
>    | ClinicalCamel-70B | +23.1  |    **+18.9**     |   +22.4    |     +23.8      |
>    | Meditron-70B      | +18.1  |    **+10.2**     |   +19.5    |     +17.9      |
>    | Med42-70B         | +20.4  |    **+14.1**     |   +20.2    |     +22.0      |
>    | Llama3-70B        | +27.8  |    **+22.3**     |   +27.5    |     +29.4      |
>    | GPT-3.5-turbo     | +17.1  |     **+9.0**     |   +18.1    |     +19.6      |
>    | GPT-4o            | +29.3  |    **+25.8**     |   +29.8    |     +30.8      |

---

> ### Comment · Reviewer_D3So · 2024-11-23
>
> Thank you for your detailed clarifications.
>
> > Our proposed PretexEval framework, on the other hand, generates test samples based on knowledge points from medical knowledge bases, which are regularly maintained by experts and publicly accessible (e.g., UMLS, DrugBank). Therefore, our framework can effectively mitigate the problem of outdated knowledge and also dynamically generate diverse samples based on medical knowledge.
>
> The clarification provided by the authors is understandable. However, my concern lies in the fact that the data is likely constructed using knowledge from fixed timepoints, such as DrugBank or UMLS. If these sources are not frequently updated, the dataset itself might risk becoming outdated.
>
> > However, since each generation uses a fixed template, the diversity of the generated knowledge expressions is limited, making it difficult to achieve a comprehensive evaluation of LLMs’ medical knowledge mastery.
>
> That said, the clarification provided is indeed very understandable. As the authors are aware, increasing the diversity of templates could potentially yield new observations, which I find to be an interesting point for further exploration.
>
> I would update my overall assessment scores due to the constructive explanations and resolving my confusing points.

---

> > ### Author Response · Authors · 2024-11-26
> > **Thank you for your response**
> >
> > We sincerely appreciate your detailed and constructive feedback.
> >
> > **Concern 1**: *The data is likely constructed using knowledge from fixed timepoints, such as DrugBank or UMLS. If these sources are not frequently updated, the dataset itself might risk becoming outdated.*
> >
> > **Answer 1**: We are very grateful for your thoughtful comments. Indeed, the infrequently updated medical knowledge bases may potentially lead to PretexEval-generated datasets becoming outdated. Nevertheless, overall, PretexEval can leverage a range of medical knowledge bases (there are plenty of high-quality knowledge bases in the medical domain) to automatically generate diverse evaluation sets, enabling a broader coverage of medical knowledge compared to existing medical benchmarks (e.g., MedQA). Moreover, the update frequency of medical knowledge bases, maintained by dedicated organizations in the medical field, is generally more timely than that of many LLM medical benchmarks. For instance, UMLS is updated semi-annually, and DrugBank is updated quarterly. This could ensure the timeliness of evaluation sets generated by PretexEval.
> >
> > **Concern 2**: *Increasing the diversity of templates could potentially yield new observations, which I find to be an interesting point for further exploration.*
> >
> > **Answer 2**: Thank you very much for your insightful suggestions. This is indeed a very interesting point. Here, we randomly sampled 200 knowledge triplets from the medical knowledge bases DiseK to conduct a very preliminary study by manually constructing multiple templates (adding another two different templates) to directly produce test samples. We primarily conducted evaluations on the LLaMA series and medical-specific LLMs, and the results (average accuracy gain compared to random guessing) are presented below:
> >
> > | Model             | Direct (single-template) | Direct (multi-template) | PretexEval |
> > | ----------------- | ------------------------ | ----------------------- | ---------- |
> > | llama2-7B         | +13.0                    | +10.1                   | +3.8       |
> > | llama3-8B         | +17.8                    | +19.0                   | +8.8       |
> > | llama2-70B        | +19.3                    | +19.3                   | +9.3       |
> > | llama3-70B        | +27.3                    | +27.1                   | +20.0      |
> > | ClinicalCamel-70B | +24.3                    | +22.1                   | +15.5      |
> > | Meditron-70B      | +19.8                    | +21.7                   | +9.0       |
> > | Med42-70B         | +22.5                    | +21.4                   | +12.8      |
> > | Average           | +20.5                    | +20.1                   | +11.3      |
> >
> > We observed that increasing template diversity generally causes a slight performance decline in most LLMs. This suggests that increasing the diversity of templates can enhance the diversity of generated samples to some extent. Nevertheless, the performance of LLMs evaluated by PretexEval remains significantly lower, further indicating that PretexEval can dynamically generate samples with greater structural and lexical diversity.

---

### Official Review · Reviewer_hmcp · 2024-11-03

**Soundness:** 3
**Presentation:** 3
**Contribution:** 3
**Rating:** 8
**Confidence:** 3

**Summary:**

The paper presents PretexEval, a novel evaluation framework designed to assess the mastery of medical knowledge by large language models (LLMs). Recognizing the limitations of existing evaluation methods, which often produce test samples with factual errors and insufficient diversity, PretexEval introduces a new schema based on predicate equivalence transformations. This approach generates a series of variant samples from a given medical knowledge point, ensuring both reliability and diversity in the test samples created. This framework was applied to evaluate 12 well-known LLMs using two medical knowledge bases. The findings reveal that, despite notable successes on standard QA benchmarks, current LLMs exhibit significant deficiencies in thoroughly understanding medical knowledge.

**Strengths:**

Originality: The concept of dynamically generating test samples from medical knowledge bases using predicate equivalence transformations is innovative.

Quality: The experimental design is robust, involving a comprehensive evaluation of 12 well-known LLMs across two distinct medical knowledge bases. The methodology for transforming predicates into diverse textual samples ensures that the evaluations are rigorous. The detailed ablation study and the use of two evaluation metrics (average accuracy and joint accuracy) further substantiate the quality of the research.

Clarity: The paper is well-organized and articulates ideas clearly and logically.

Significance: The findings highlight significant deficiencies in the current LLMs' ability to fully master medical knowledge, which is critical for their deployment in real-world medical scenarios.

**Weaknesses:**

1) The limitations of the proposed approach should be discussed in the paper:
- The PretexEval framework heavily relies on predicate transformations to generate test samples. While this approach contributes to sample diversity, it may not adequately capture the complexity or the nuances of medical reasoning that goes beyond simple factual recall.
- The current evaluation metrics, while useful, focus predominantly on binary true/false assessments of knowledge mastery. This binary approach might oversimplify the evaluation of medical knowledge, where a more nuanced understanding might be necessary.

2) The code and datasets should be made publicly available for reproducibility purposes.

**Questions:**

- In comparison with the previous subsections, section 3.2.2 is harder to follow. To facilitate understanding, this section would greatly benefit from including examples (particularly lines 211-213).

---

> ### Author Response · Authors · 2024-11-22
> **Response to Reviewer hmcp**
>
> We sincerely appreciate your detailed and constructive feedback. Below, we provide our responses to each of the concerns you have raised.
>
> 1. **Limitations of PretexEval**: Thank you for your constructive feedback. Although PretexEval can dynamically generate diverse and reliable samples based on medical knowledge bases, it also has the following two limitations: (1) While PretexEval can generate diverse samples and incorporate different tasks to evaluate LLMs’ capabilities in medical knowledge application, for extremely complex medical tasks, the samples generated by PretexEval may not well-suited for integration with these tasks; (2) Although PretexEval effectively generate diverse and reliable samples, to ensure the readability of the generated samples, we leverage LLM rephrasing during the sample generation process, which may potentially introduce some uncertainty. We have provided a more detailed discussion of these limitations in the conclusion part of our revised paper.
>
> 2. **Current evaluation metrics focus predominantly on binary true/false assessments**: Thank you for your constructive suggestions. We primarily used true/false verification as the evaluation metric, since the generated knowledge expressions can be easily converted into this format. In fact, apart from the true/false verification task, PretexEval is highly scalable and can further integrate with various evaluation tasks. To validate the scalability of PretexEval, we conduct a preliminary study by further incorporating PretexEval with multiple-choice question (MCQ) tasks. Specifically, we first randomly selected 200 knowledge triplets from one of the knowledge bases utilized in this study (MedLAMA knowledge base). Next, we generate 1,600 reliable and diverse samples using PretexEval, and further rephrase them into the corresponding MCQ questions. The experimental results (average accuracy gain compared to random guessing) are listed below:
>
>    | Models            | LLMEval | PretexEval (Ours) |
>    | ----------------- | :-----: | :---------------: |
>    | Llama2-7B         |  +9.4   |     **+2.1**      |
>    | Vicuna-7B         |  +25.7  |     **+12.1**     |
>    | Vicuna-13B        |  +36.3  |     **+12.5**     |
>    | Gemma-7B          |  +57.6  |     **+26.7**     |
>    | Llama3-8B         |  +55.5  |     **+32.2**     |
>    | Llama2-70B        |  +62.9  |     **+35.8**     |
>    | ClinicalCamel-70B |  +64.0  |     **+36.0**     |
>    | Meditron-70B      |  +62.6  |     **+30.2**     |
>    | Med42-70B         |  +66.3  |     **+39.1**     |
>    | Llama3-70B        |  +66.6  |     **+42.7**     |
>    | GPT-3.5-turbo     |  +38.6  |     **+16.7**     |
>    | GPT-4o            |  +67.6  |     **+49.8**     |
>
>    We observe a strong correlation between LLMs' performance on MCQ questions and their performance on true/false verification tasks presented in our paper (Spearman rank correlation coefficient = 0.91, p-value = 4e-05). Moreover, the evaluated LLMs consistently achieve much lower performance on samples generated by PretexEval compared to those generated by LLMEval, reflecting a trend similar to the results reported in our paper. The experimental results above suggest that PretexEval can also produce reasonable evaluation outcomes when integrated with the MCQ task. We provide detailed results of this preliminary study in Figure 7 of our revised paper. Though PretexEval can theoretically be integrated with various evaluation tasks, evaluating LLMs’ more nuanced understanding of medical knowledge would certainly require a more refined design of the evaluation framework.
>
> 3. **Code & Data Avaliability**: Thank you for your kind suggestions. We will make all the codes and datasets used in this study publicly available to ensure reproducibility and support further research in this area.
>
> 4. **Section 3.2.2 is harder to follow**: We sincerely appreciate your detailed and constructive suggestions. Following your suggestions, we have included examples in this section and revised the relevant wording to improve the clarity of this subsection.

---

### Official Review · Reviewer_jTNo · 2024-11-08

**Soundness:** 3
**Presentation:** 3
**Contribution:** 3
**Rating:** 8
**Confidence:** 4

**Summary:**

PretexEval is a new way to assess how well large language models (LLMs) understand medical knowledge by creating dynamic test samples from medical databases. The framework uses simple logic transformations (like flipping statements, modifying them, and using double negatives) to produce a variety of reliable test questions, going beyond traditional fixed tests. The authors evaluate 12 well-known LLMs, including some designed specifically for medicine, using two databases focused on clinical diagnosis and treatment. They show that while these models perform well on existing medical benchmarks, they still struggle to consistently grasp medical knowledge when it’s presented in different ways. The main contribution is a flexible method for testing LLMs’ understanding of medical knowledge, backed by detailed results that reveal significant gaps between their performance on tests and their genuine understanding of medical concepts. This research offers further insights into methods for setting up a systematic way to scale the evaluation of LLM readiness for use in clinical settings.

**Strengths:**

- PretexEval introduces a new approach to testing medical knowledge in language models. Rather than using fixed test questions, it dynamically generates diverse test cases that probe deeper understanding. This represents a significant methodological advancement in AI evaluation.
- The authors create a reliable yet flexible testing framework by combining predicate transformations with natural language generation. The method is both principled and practical, with clear steps for reproduction.
- The empirical results reveal important insights about current AI capabilities in medicine. The authors demonstrate that even state-of-the-art models struggle with consistent understanding by testing 12 different models with varying forms of the same medical knowledge. This finding has direct implications for clinical applications.
- The framework's generalizability is a key strength. While demonstrated on two medical knowledge bases, the approach can extend to other medical domains and potentially beyond medicine. This broad applicability makes the contribution particularly valuable for the field.
- LLMs across sizes and even API based models were evaluated and compared against simple LLM rephrasing

**Weaknesses:**

The study's methodology, while strong, has a few areas that could be strengthened.
- Though clearly explained for simple relationships, the predicate transformation concept lacks a formal specification for handling complex medical relationships that don't cleanly map to simple predicates.
- Further, the evaluation is limited to one relationship test in a multiple-choice setting, which may not fully capture the complexity of medical knowledge application. Exploring the handling of multiple variations, such as incorporating distractors or relationship swapping similar to GSM symbolic approaches, could provide a more comprehensive assessment of LLM capabilities in medical contexts.
- Currently, the framework is only tested for binary settings; it would be useful to know how this setting can be applied to even a simple 4-class mcq or, ideally, an open-ended setting.

**Questions:**

In the case study shown in Figure 6, the PretexEval method includes a rewording that shifts from "may treat" to "is recommended." This raises concerns about whether this transformation validly rephrases the original medical relationship. In medical contexts, "may treat" and "is recommended" carry different implications:
"May treat" suggests a possibility or potential for treatment, leaving room for other options or uncertainties.
"Is recommended" implies a stronger clinical endorsement, suggesting that the treatment is advised or endorsed by clinical guidelines or experts.
Given this distinction, the shift from "may treat" to "is recommended" could be problematic if it alters the strength of the claim. This transformation might lead to incorrect evaluations, as LLMs could be unfairly penalized for failing to recognize a relationship that has been shifted beyond its original meaning.
Therefore an important question arises:
- Clarification on Predicate Transformation Boundaries: Could you clarify how you ensure that predicate transformations do not alter the underlying meaning of medical relationships? Specifically, how do you prevent shifts in meaning from "may treat" (a possibility) to "is recommended" (a stronger clinical assertion)? I understand some validation in the initial setting was completed, but this may be lost during rephrasing- so I just want to clarify what of the final questions were validated and by what grade physicians.
- Handling Ambiguities in Medical Relationships: How does PretexEval handle nuanced medical relationships where different levels of certainty or recommendation are involved? For example, how do you ensure that transformations like "may treat" vs. "is recommended" retain their original intent without introducing stronger or weaker claims?
- Validation of Rephrased Statements: Could you describe any validation steps taken by medical experts to confirm that reworded statements retain their original intent and do not introduce factual inaccuracies or misleading implications?
- Addressing these concerns would help ensure that the transformations generated by PretexEval maintain clinical accuracy and do not inadvertently alter the meaning of medical knowledge points during evaluation.

---

> ### Author Response · Authors · 2024-11-22
> **Response to Reviewer jTNo (1/3)**
>
> We would like to express our gratitude for your thoughtful and constructive feedback. Below are our responses to each of the concerns you raised.
>
> 1. **Handling complex medical relationships**: We sincerely appreciate your thoughtful comments. Generally, complex multi-hop medical relationships can be broken down into multiple single-hop relationships, which can then be mapped to corresponding predicates. As a result, we can describe this multi-hop relationship using the combination of these predicates (i.e., composite predicates), and then evaluate this complex medical relationship through PretexEval. For example, for the complex medical relationship *Aspirin is used to treat heart disease, but long-term use may cause stomach bleeding,* we can decompose this into two single-hop relationships: *(Aspirin, treats, Heart Disease)* and *(Aspirin, may cause, Stomach Bleeding)*. Further, through predicate transformations (e.g., Inversion), we can generate diverse expressions based on this knowledge, such as *Heart disease can be treated with aspirin, while stomach bleeding may be caused by the long-term use of aspirin*. Indeed, as you mentioned, some types of complex medical relationships may remain challenging to describe using predicates. We plan to explore new evaluation methods for these types of medical knowledge in the future.
>
> 2. **Applying PretexEval to other test types for capturing the complexity of medical knowledge application**: We are very grateful for your constructive suggestions. In this work, we primarily leverage the test setting of true/false verification in PretexEval since the generate knowledge expressions can be conveniently transformed into statement verification questions. Nonetheless, it is worth highlighting that the proposed PretexEval framework is **highly scalable** in terms of evaluation tasks and is compatible with various task settings, including the 4-class multiple-choice questions (MCQs) you kindly mentioned. This scalability arises from the framework’s ability of generating diverse and reliable expressions based on medical knowledge, which can be further integrated with different test settings. To validate the scalability of PretexEval, we follow your kind advice conduct a preliminary study by further incorporating PretexEval with 4-option multiple-choice question (MCQ) tasks. Specifically, we first randomly selected 200 knowledge triplets from one of the knowledge bases utilized in this study (MedLAMA knowledge base). Next, we generate 1,600 reliable and diverse samples using PretexEval, and further rephrase them into the corresponding MCQ questions. The experimental results (average accuracy gain compared to random guessing) are listed below:
>
>    | Models            | LLMEval | PretexEval (Ours) |
>    | ----------------- | :-----: | :---------------: |
>    | Llama2-7B         |  +9.4   |     **+2.1**      |
>    | Vicuna-7B         |  +25.7  |     **+12.1**     |
>    | Vicuna-13B        |  +36.3  |     **+12.5**     |
>    | Gemma-7B          |  +57.6  |     **+26.7**     |
>    | Llama3-8B         |  +55.5  |     **+32.2**     |
>    | Llama2-70B        |  +62.9  |     **+35.8**     |
>    | ClinicalCamel-70B |  +64.0  |     **+36.0**     |
>    | Meditron-70B      |  +62.6  |     **+30.2**     |
>    | Med42-70B         |  +66.3  |     **+39.1**     |
>    | Llama3-70B        |  +66.6  |     **+42.7**     |
>    | GPT-3.5-turbo     |  +38.6  |     **+16.7**     |
>    | GPT-4o            |  +67.6  |     **+49.8**     |
>
> We observe a strong correlation between LLMs' performance on MCQ questions and their performance on true/false verification tasks presented in our paper (Spearman rank correlation coefficient = 0.91, p-value = 4e-05). Moreover, the evaluated LLMs consistently achieve much lower performance on samples generated by PretexEval compared to those generated by LLMEval, reflecting a trend similar to the results reported in our paper. The experimental results above suggests that PretexEval can generate more diverse samples across different task settings to better evaluate LLMs' mastery of medical knowledge. The detailed results of this preliminary study is provided in Figure 7 of our revised paper.
>
> Thank you again for your constructive feedback. PretexEval can theoretically be combined with various test types (multiple-answer questions, open-ended questions) to fully capture the complexity of medical knowledge application. In the future, we also plan to further construct a larger medical knowledge base and integrate it with various evaluation tasks to create a more comprehensive medical evaluation framework.

---

> > ### Author Response · Authors · 2024-11-22
> > **Response to Reviewer jTNo (2/3)**
> >
> > 3. **Clarification on Predicate Transformation Boundaries**: We greatly appreciate your insightful and professional comments regarding the reliability of expressions generated by PretexEval. Ensuring that the generated samples faithfully reflect the original intent of medical knowledge, including the *strength of claims* you mentioned, is indeed very important and a key focus of our research. In this study, we initially attempted to directly prompt LLMs to convert predicate variants into textual expressions. However, the results of clinical expert validation revealed insufficient reliability issues with this method. For example, given the predicate *morphology_of(Internal traumatic fistula, Urethral false passage)*, the LLM-converted expression *"The morphology associated with internal traumatic fistula is that of a urethral false passage"* mistakenly inverted the head and tail entities, leading to an incorrect expression. To address this, we designed a prototype-based method. We first use prototype templates to convert predicate variants into textual expressions and then utilized LLMs to rephrase these textual expressions, enhancing both readability and lexical diversity. We carefully designed these prototype templates to ensure that they faithfully preserve the original meaning of corresponding relationships, including the *strength of claims*. Expert evaluations demonstrated that the knowledge expressions before LLM rephrasing exhibited high reliability (as shown by the yellow bar in Figure 5 of our paper). However, while the LLM rephrasing process effectively improved lexical diversity, it caused a slight (~0.16) reduction in reliability, as reflected in the brown bar in Figure 5. By carefully reviewing the results of clinical expert validation, we found that the *strength of claims* in a small portion of samples was changed by the LLM rephrasing process, which could be one of the reasons for this reduction in reliability.
> >
> > We further conduct an additional experiment to study how the *strength of claims* affect the performance of LLMs evaluated by our framework. Specifically, we consider rewording the original knowledge expressions in two ways:
> >
> > + Rewording the expression without changing the strength of claim (**w/o strength change**)
> > + Rewording the expression with the strength of claim changed (**w strength change**)
> >
> > The formats of constructed expressions are categorized and listed below.
> >
> > | Original                   | w/o strength change                             | w/ strength change                                   |
> > | -------------------------- | ----------------------------------------------- | ---------------------------------------------------- |
> > | Drug A may treat Disease B | Drug A may be used to treat Disease B           | Drug A can be useful for treating Disease B          |
> > |                            | Drug A has the potential for treating Disease B | Drug A is recommended for the treatment of Disease B |
> >
> > Subsequently, we utilize these expressions to generate test samples based on 100 randomly sampled knowledge triplets. We evaluate several typical LLMs using the generated samples, with the experimental results (average accuracy) listed below:
> >
> > | Model       | Original | w/o strength change | w/ strength change |
> > | ----------- | :------: | :-----------------: | :----------------: |
> > | llama2-7B   |   70.0   |        67.5         |        68.8        |
> > | llama3-8B   |   84.5   |        83.0         |        82.7        |
> > | llama2-70B  |   85.0   |        84.7         |        83.3        |
> > | llama3-70B  |   90.5   |        91.0         |        91.3        |
> > | GPT-4o      |   86.0   |        86.3         |        86.0        |
> > | **Average** |   83.2   |        82.5         |        82.4        |
> >
> >  We found that LLMs generally perform slightly worse on reworded expressions compared to the original expressions, regardless of variations in claim strength. Moreover, on reworded expressions, LLMs perform similarly across different levels of claim strength. This suggests that while rewording slightly impacts the performance of LLMs, variations in claim strength have minimal effect to the performance of LLMs. Therefore, the conclusions in our paper are still valid.
> >
> > Balancing the diversity and reliability of generated samples is indeed a challenging problem. Inspired by your valuable comments, we plan to further enhance the reliability of generated samples by adding a new module that automatically detects potential changes in claim strength and keeps the original strength of claim unchanged in such situations in the future.

---

> > > ### Author Response · Authors · 2024-11-22
> > > **Response to Reviewer jTNo (3/3)**
> > >
> > > 4. **Validation of Rephrased Statements**: We are very thankful for your kind comments. In fact, in this work, we initially engaged two medical experts (both licensed physicians with 3–6 years of experience) to validate a total of 150 medical knowledge expressions generated by three different methods. The methods for generating samples include:
> > >
> > > + LLMEval: Generating diverse knowledge expressions directly from medical knowledge through prompting LLMs.
> > > + PretexEval w/o Rephrasing: Generating expressions based on predicate and prototype-based transformations, without LLM rephrasing.
> > > + PretexEval：Further enhancing readability and lexical diversity of expressions derived from the previous step through LLM rephrasing.
> > >
> > > The medical experts validated the generated expressions by comparing them with the original medical knowledge triplets, regarding their reliability, lexical diversity, and structural diversity. For reliability, each expert is asked to carefully compare the given expression with the original medical knowledge triple to determine whether the generated expression accurately preserves the original meaning without introducing factual errors, and to assign a score ranging from 0 to 5. Based on the suggestion from Reviewer fVf4, we further engaged an additional senior medical expert (with about 8 years of experience) for validation and measured the inter-annotator agreement (using Intraclass Correlation Coefficient (ICC), where an ICC > 0.9 indicates excellent agreement) across all the annotators. We present the inter-annotator agreement metrics and corresponding 95% confidence intervals below. We found that the validation results change marginally after engaging in this new expert. Moreover, the validation results from the three experts demonstrated strong consistency, with inter-annotator agreement exceeding 0.9 (see table below), highlighting the robustness of our clinical expert validation process. We have included the updated validation results in Figure 5 and refined the corresponding descriptions to enhance clarity in our revised paper.
> > >
> > > | Dimensions           | Inter-annotator Agreement Coefficient | Confidence Interval |
> > > | -------------------- | :-----------------------------------: | :-----------------: |
> > > | Reliability          |                 0.912                 |    [0.88, 0.93]     |
> > > | Lexical Diversity    |                 0.938                 |     [0.92,0.95]     |
> > > | Structural Diversity |                 0.956                 |     [0.94,0.97]     |

---

> > > > ### Comment · Reviewer_jTNo · 2024-11-28
> > > > **Response**
> > > >
> > > > Thank you for your detailed reply and the extra results you have shown. Given the response i have updated my score.

---

### Official Review · Reviewer_fVf4 · 2024-11-09

**Soundness:** 3
**Presentation:** 2
**Contribution:** 3
**Rating:** 6
**Confidence:** 4

**Summary:**

The paper presents PretexEval, an innovative evaluation framework designed to assess medical LLMs' knowledge mastery with a focus on generating reliable, diverse test samples. By transforming medical knowledge points through predicate equivalence transformations, PretexEval produces varied yet accurate textual representations, allowing for a richer evaluation compared to traditional benchmarks like MedQA. The framework not only highlights areas where LLMs are deficient but also introduces joint accuracy as a critical metric to measure the consistency of LLMs’ responses—an important yet often overlooked requirement in healthcare settings.

**Strengths:**

- Novel Evaluation Method: PretexEval introduces a fresh approach by transforming knowledge points into diverse predicates, effectively improving both the reliability and comprehensiveness of LLM evaluation. This is particularly relevant in fields like healthcare, where factual consistency is crucial.
- Focus on Consistency: The framework’s use of joint accuracy to measure consistency across expressions of the same knowledge point is a valuable contribution. In healthcare, where consistency is key, this metric allows for a deeper look at whether a model can reliably interpret nuanced variations of medical knowledge.
- Broad Evaluation Scope: The study evaluates 12 leading LLMs across both general and medical-specific models, demonstrating PretexEval’s versatility and highlighting critical insights into current LLM limitations.

**Weaknesses:**

- Scope of Evaluation Tasks: The framework is limited to true/false verification tasks, which could constrain its applicability in complex medical scenarios where contextual understanding and reasoning are required.
- Prototype Dependence: Manually crafted prototypes, while improving reliability, may limit scalability and introduce potential subjectivity. Refining this step or automating parts of it could enhance PretexEval’s flexibility. Reliability here is measured by two annotators on a 50-sample subset, introducing a skewed potential for bias. Expanding this evaluation to at least three annotators and including an inter-annotator agreement measure would add robustness.
- Applicability for Model Training: While it’s a promising evaluation tool, exploring the potential of PretexEval-generated samples for training might add another layer of impact, possibly making it a resource for enhancing models’ medical knowledge consistency.

**Questions:**

- How do you plan to extend PretexEval beyond true/false verification tasks? Adding context-sensitive or reasoning-based tasks could expand its applicability to more complex clinical scenarios.

- Prototype Automation: Are there plans to automate prototype creation to reduce potential subjectivity and improve scalability? If not, what criteria ensure prototype reliability and consistency across varied medical contexts?

- Reliability Validation: Given the small sample size and limited number of annotators for reliability measurement, how would expanding the annotation set or incorporating inter-annotator agreement metrics influence reliability scores? Would this improve the robustness of results?

- Could PretexEval-generated samples serve a role in training, not just evaluation? If incorporated into training, would these samples likely boost LLMs’ consistency on diverse medical knowledge?

---

> ### Author Response · Authors · 2024-11-22
> **Response to Reviewer fVf4 (1/3)**
>
> 1. **Extending PretexEval Beyond True/false Verification Tasks**：We are sincerely grateful for your constructive comments. In this work, we primarily adopt true/false verification task in PretexEval since the generate knowledge expressions can be directly transformed into statement verification questions. However, it is worth mentioning the proposed PretexEval framework is inherently **highly scalable** and can be adapted to various task forms (e.g., multiple-choice question-answering). This scalability arises from the framework’s core capability of generating diverse and reliable expressions based on medical knowledge points, which can be further integrated with different types of evaluation tasks. Thank you again for your constructive feedback. To validate the scalability of PretexEval, we conduct a preliminary study by further incorporating PretexEval with multiple-choice question (MCQ) tasks. Specifically, we first randomly selected 200 knowledge triplets from one of the knowledge bases utilized in this study (MedLAMA knowledge base). Next, we generate 1,600 reliable and diverse samples using PretexEval, and further rephrase them into the corresponding MCQ questions. The experimental results (average accuracy gain compared to random guessing) are listed below:
>
>    | Models            | LLMEval | PretexEval (Ours) |
>    | ----------------- | :-----: | :---------------: |
>    | Llama2-7B         |  +9.4   |     **+2.1**      |
>    | Vicuna-7B         |  +25.7  |     **+12.1**     |
>    | Vicuna-13B        |  +36.3  |     **+12.5**     |
>    | Gemma-7B          |  +57.6  |     **+26.7**     |
>    | Llama3-8B         |  +55.5  |     **+32.2**     |
>    | Llama2-70B        |  +62.9  |     **+35.8**     |
>    | ClinicalCamel-70B |  +64.0  |     **+36.0**     |
>    | Meditron-70B      |  +62.6  |     **+30.2**     |
>    | Med42-70B         |  +66.3  |     **+39.1**     |
>    | Llama3-70B        |  +66.6  |     **+42.7**     |
>    | GPT-3.5-turbo     |  +38.6  |     **+16.7**     |
>    | GPT-4o            |  +67.6  |     **+49.8**     |
>
> We observe a strong correlation between LLMs' performance on MCQ questions and their performance on true/false verification tasks presented in our paper (Spearman rank correlation coefficient = 0.91, p-value = 4e-05). Moreover, the evaluated LLMs consistently achieve much lower performance on samples generated by PretexEval compared to those generated by LLMEval, reflecting a trend similar to the results reported in our paper. The experimental results above suggests that PretexEval can generate more diverse samples across different task settings to better evaluate LLMs' mastery of medical knowledge. The detailed results of this preliminary study is provided in Figure 7 of our revised paper.
>
> In summary, PretexEval can generate diverse and reliable samples based on medical knowledge, which can theoretically be incorporated with various types of tasks to evaluate LLMs’ contextual understanding and reasoning abilities related to the corresponding medical knowledge. We plan to extend PretexEval to support a broader range of evaluation tasks in the future.

---

> > ### Author Response · Authors · 2024-11-22
> > **Response to Reviewer fVf4 (2/3)**
> >
> > 2. **Prototype Automation**: We sincerely appreciate your thoughtful comments. In this work, we leverage prototypes to balance PretexEval’s reliability and scalability in converting predicate variants into textual expressions. Manually constructing textual expressions for each predicate variant of every knowledge triplet ensures reliability but lacks scalability and introduces potential subjectivity. Conversely, directly using LLMs to convert predicate variants into textual expressions offers strong scalability but may potentially introduces factual errors during generation, resulting in lower reliability. Our proposed prototype-based method, on the other hand, serves as a semi-automated solution: for each combination of predicate transformation type and relationship type in the knowledge base, we manually design a prototype and then use it to automatically convert predicate variants into corresponding textual expressions. For example, for the relationship type *may_treat* combined with the *Inversion* transformation type, we construct the following prototype:
> >
> > ```
> > [Tail Entity] may be treated by using [Head Entity]
> > ```
> >
> > Leveraging this prototype, we can generate corresponding textual expressions for all knowledge triplets of the same relation type. For example, for the triplet *(riboflavin tetrabutyrate, may treat, riboflavin deficiency)*, we replace the placeholders in the prototype with the head and tail entities, resulting in the corresponding knowledge expression:
> >
> > ```
> > [Tail Entity] may be treated by using [Head Entity] -> Riboflavin deficiency may be treated by using riboflavin tetrabutyrate.
> > ```
> >
> > Considering the limited number of relation types in medical knowledge bases, the manual construction of prototypes requires relatively little effort and does not scale with the number of triplets in the knowledge base. Additionally, the prototypes we construct are relatively simple, involving minimal subjectivity. By further rephrasing the generated knowledge expressions using an LLM, we can improve the readability and lexical diversity of the expressions while maintaining reliability. Thank you again for your valuable suggestion. We also plan to explore methods for automating prototype construction in the future, while such automated prototypes may also require thorough analysis to ensure their reliability.
> >
> > 3. **Clinical Expert Validation Issues**: We would like to express our gratitude to your detailed and constructive feedback. In this work, we conducted a clinical expert validation to demonstrate that PretexEval can dynamically generate diverse and reliable knowledge expressions based on medical knowledge. Specifically, we randomly selected 50 medical knowledge triples and engaged two medical experts (both licensed physicians with 3–6 years of experience) to manually evaluate the reliability, lexical diversity, and structural diversity of 150 medical knowledge expressions generated by three different methods. Based on your valuable suggestions, we further engaged a **senior medical expert** (with about 8 years of experience) and selected the Intraclass Correlation Coefficient (ICC), specifically the ICC2k (Average random raters) type, as the metric to measure the inter-annotator agreement. The inter-annotator agreement coefficients, along with the 95% confidence intervals, are provided in the table below. We found that the manual validation results changed minimally after involving the new expert, and the inter-annotator agreement scores across all three evaluation dimensions exceeded 0.9, demonstrating **excellent consistency among the three experts**. The validation results demonstrate that PretexEval-generated samples ensure reliability while achieving considerable diversity. This results of clinical expert validation has also been updated in our paper (Figure 5).
> >
> > | Dimensions           | Inter-annotator Agreement Coefficient | Confidence Interval |
> > | -------------------- | :-----------------------------------: | :-----------------: |
> > | Reliability          |                 0.912                 |    [0.88, 0.93]     |
> > | Lexical Diversity    |                 0.938                 |     [0.92,0.95]     |
> > | Structural Diversity |                 0.956                 |     [0.94,0.97]     |

---

> ### Author Response · Authors · 2024-11-22
> **Response to Reviewer fVf4 (3/3)**
>
> 4. **Applicability for Model Training**: Thank you for your constructive suggestions. This study primarily aims to evaluate LLMs' mastery of medical knowledge. Your feedback is valuable, highlighting the potential of PretexEval-generated samples for training to enhance LLMs’ medical knowledge consistency. Based on your kind suggestion, we have conducted a preliminary study to explore the potential of training with PretexEval-generated samples. We proposed two research questions for investigation:
>
>   + **RQ1:** Can training on PretexEval-generated samples improve LLMs’ understanding of knowledge outside the training set?
>
>   + **RQ2:** Can training on a few types of PretexEval-generated samples improve LLMs’ understanding of other unseen expressions?
>
> For **RQ1**, we selected 200 knowledge triples as the training set and another 200 triples as the test set. We finetuned Llama3-8B using all PretexEval-generated samples derived from the training set, and apply PretexEval for evaluation on the test set. The experimental results (average accuracy) are reported in the table below:
>
> | Before  Training | After Training | $\Delta$ |
> | :--------------: | :------------: | :------: |
> |       66.9       |      88.8      |  +21.9   |
>
> We observe that the LLM's performance improves largely (~20%) after training on PretexEval-generated samples, **indicating that training on PretexEval-generated samples could improve LLMs’ mastery of knowledge outside the training set.**
>
> For **RQ2**, since PretexEval can generate 8 different types of expressions for a given knowledge point, we randomly selected 4 out of 8 types for training and used the remaining 4 types for evaluation. We report the average accuracy on the four evaluated types of expressions in the table below:
>
> | Before  Training | After Training | $\Delta$ |
> | :--------------: | :------------: | :------: |
> |       63.4       |      95.5      |  +32.1   |
>
>  Experimental results show that training on a subset of PretexEval-generated expressions largely improves LLMs' performance (~30%) on the unseen types of expressions. This demonstrates that **training on a few types of PretexEval-generated expressions can improve LLMs’ ability to understand unseen expressions related to the same knowledge.**
>
> We have updated our paper to include the detailed experimental results of this preliminary study in Figure 8. While the results show promise in enhancing LLMs’ medical knowledge consistency, further research is needed to make this approach practical. We leave this for future work.

---

### Meta-Review · Area_Chair_5oLZ · 2024-12-18

**Metareview:**

This paper is about evaluating the medical knowledge that is implicitly stored in LLMs. To this end, the authors introduce Predicate-to-text Evaluation, or PretexEval; this is an evaluation framework intended to robustly evaluate LLMs "mastery" of an arbitrary knowledge base. In particular, PretexEval works by generating variants of predicates derived from a knowledge base in a way that preserves their accuracy while offering diversity in test samples.

There was consensus amongst reviewers that this is a welcome contribution in the space of (medical) LLM evaluation. The approach addresses some of the weaknesses of existing static benchmarks like MedQA, and the authors provide a relatively comprehensive comparison of SOTA LLMs using PretexEval. There are some points that could use clarifying—e.g., around the transformations for complex relationships, as raised by jTNo. But as is this is a strong contribution to the critical area of evaluation; it may be impactful given the flexibility of the approach.

**Additional Comments On Reviewer Discussion:**

The authors did a commendable job of responding to reviews.

---

### Decision · Program_Chairs · 2025-01-22

Accept (Poster)